# Cell fate in antiviral response arises in the crosstalk of IRF, NF-κB and JAK/STAT pathways

Maciej Czerkies[1], Zbigniew Korwek[1], Wiktor Prus[1], Marek Kochańczyk [1], Joanna Jaruszewicz-Błońska[1], Karolina Tudelska[1], Sławomir Błoński[1], Marek Kimmel[2,3], Allan R. Brasier [4] & Tomasz Lipniacki [1]

The innate immune system processes pathogen-induced signals into cell fate decisions. How information is turned to decision remains unknown. By combining stochastic mathematical modelling and experimentation, we demonstrate that feedback interactions between the IRF3, NF-κB and STAT pathways lead to switch-like responses to a viral analogue, poly(I:C), in contrast to pulse-like responses to bacterial LPS. Poly(I:C) activates both IRF3 and NF-κB, a requirement for induction of IFNβ expression. Autocrine IFNβ initiates a JAK/STAT-mediated positive-feedback stabilising nuclear IRF3 and NF-κB in first responder cells. Paracrine IFNβ, in turn, sensitises second responder cells through a JAK/STAT-mediated positive feedforward pathway that upregulates the positive-feedback components: RIG-I, PKR and OAS1A. In these sensitised cells, the 'live-or-die' decision phase following poly(I:C) exposure is shorter—they rapidly produce antiviral responses and commit to apoptosis. The interlinked positive feedback and feedforward signalling is key for coordinating cell fate decisions in cellular populations restricting pathogen spread.

[1] Institute of Fundamental Technological Research, Polish Academy of Sciences, Warsaw 02-106, Poland. [2] Departments of Statistics and Bioengineering, and Program in Systems, Synthetic, and Physical Biology, Rice University, Houston, TX 77005, USA. [3] Systems Engineering Group, Silesian University of Technology, Gliwice 44-100, Poland. [4] Institute for Translational Sciences, University of Texas Medical Branch, Galveston, TX 77555-1060, USA. Maciej Czerkies, Zbigniew Korwek, Wiktor Prus, and Marek Kochańczyk contributed equally to this work. Correspondence and requests for materials should be addressed to T.L. (email: tlipnia@ippt.pan.pl)

Molecular networks process analogue signals into discrete cell fate decisions[1]. Information processing employs regulatory elements such as gene switches, logic gates, or feedback/feedforward loops[2]. In the NF-κB pathway, negative feedbacks mediated by NF-κB inhibitors, IκBα and A20, transform tonic TNFα[3,4], IL1[5] or LPS[6–8] signals into oscillatory or pulse-like responses. Positive feedbacks may lead to bi- or multistability allowing cells to assume one of mutually exclusive states depending on the strength and/or duration of stimuli[9,10]. Interlinked negative and positive feedbacks may lead to a more elaborate behaviour, that combines oscillatory responses with binary switches[11]. Pathways that evolved to respond to stress are governed by systems of coupled feedbacks[12] that may also involve cell-to-cell communication[13]. The question is how the specific topologies of these networks enable cell fate decisions. Here, to address this question we combine mathematical modelling and experimental validation, and analyse how feedbacks coupling NF-κB, IRF3 and STAT pathways govern the innate immune system and drive cells into the antiviral state and apoptosis.

Even though bacterial LPS and a viral nucleic acid analogue, poly(I:C), activate the same innate immunity pathways, the response characteristics are stimulus-dependent[14]. LPS elicits transient or oscillatory activation of NF-κB, terminated by synthesis of IκBα and A20[6–8]. The response to poly(I:C) has different dynamics. Most cells are inert, but a fraction respond by stable activation of IRF3, NF-κB and STAT1/2, and eventually commit to apoptosis. Cell fate is not determined exclusively by the stimuli but also depends on the initial state of the cell (extrinsic noise) and stochasticity in signal processing (intrinsic noise)[15,16]. Higher organisms with intercellular signalling may benefit from stochasticity by keeping only a subpopulation of cells sensitive to particular stimuli. Recent research demonstrated the role of stochasticity-driven population heterogeneity and paracrine signal propagation in shaping the antiviral response of cell population[17–19]. Here we investigate the interconnections of the major signalling arms of the innate immune response to viral patterns schematically shown in (Fig. 1a). We identify autocrine and paracrine feedbacks coupling the IRF3, NF-κB and STAT1/2 pathways, that allow for proportionate cell fate decisions coordinated across heterogeneous populations. Our data suggest that a small population of the sensitive cells form the first line of defence and sensitise other cells by secreting IFNβ. The IFNβ-primed cells

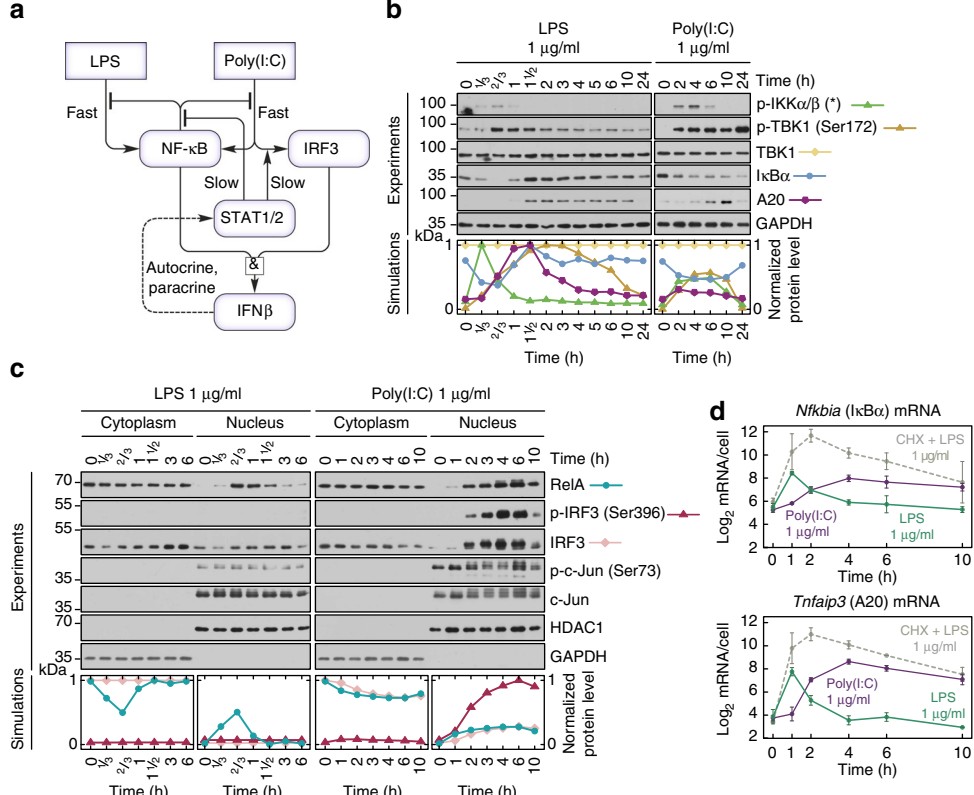

Fig. 1 LPS and poly(I:C) elicit divergent responses. a Schematic diagram of the regulatory system of three transcription factors, NF-κB, IRF3 and STAT1/2, activated upon stimulation with LPS or poly(I:C). The synthesis of cytokine IFNβ, mediating autocrine and paracrine signalling, requires activation of both NF-κB and IRF3. Arrow heads = activation, hammer heads = inhibition. b, c Protein levels of the system components in response to LPS or poly(I:C), characterised by western blotting and compared with numerical model simulations. WT MEFs were stimulated with 1 μg/ml LPS or 1 μg/ml poly(I:C). GAPDH and HDAC1 serve as loading controls. Trajectories show averages of 200 independent stochastic simulations; the colour key is located next to protein labels. b Whole-cell extracts were analysed using antibodies against phosphorylated (active) forms of IKKα/β and TBK1, as well as total TBK1, IκBα and A20. Representative experiments out of 2 for LPS and 4 for poly(I:C) are shown. (*) = IKK isoform-dependent phosphorylation sites: p-IKKα Ser176/180, p-IKKβ Ser177/181. c Cytoplasmic and nuclear fractions were analysed using antibodies against total RelA (NF-κB), IRF3 and c-Jun, as well as for phospho-forms (active forms) of IRF3 and c-Jun. Representative experiments out of 2 are shown. d mRNA levels of NF-κB inhibitors, IκBα and A20, in response to LPS, cycloheximide (CHX) with LPS, or poly(I:C). WT MEFs were stimulated with 1 μg/ml LPS in the absence or presence of 5 μg/ml CHX, or with 1 μg/ml poly(I:C). CHX was added 1 h prior to LPS stimulation starting at time = 0. Time profiles of relative mRNA levels were obtained with RT-PCR and then rescaled to absolute numbers using digital PCR measurements. Bars represent means ± s.e.m., n ≥ 2, see Supplementary Note for plots of all replicates compared with model simulations

have increased levels of positive-feedback components, which allows them to shorten the 'live-or-die' decision phase and increase their apoptotic rate after a subsequent poly(I:C) stimulation. In the following, we discuss the data gathered to derive the mathematical model of innate immune responses. For brevity, even before presenting the model, we juxtapose experimental and simulated protein time profiles.

## Results

**LPS and poly(I:C) elicit different NF-κB/IRF3 kinetics**. First, we characterise dynamical differences between responses to LPS and poly(I:C). In mouse embryonic fibroblasts (MEFs) both stimuli activate NF-κB with distinct kinetics (Fig. 1b, c), and only poly(I:C) leads to IRF3 activation (Fig. 1c). Interestingly, LPS activates TBK1 but not IRF3, a pattern previously reported in TRIF-deficient macrophages[20]. This suggests that in MEFs, LPS signalling is transmitted by the MyD88 pathway rather than the TRIF pathway dependent on TLR4 endocytosis[21–23]. LPS causes a rapid but transient phosphorylation of the canonical kinases IKK, followed by degradation of IκBα (Fig. 1b). This releases NF-κB, enabling its rapid nuclear translocation, with a peak at 40 min after LPS stimulation (Fig. 1c shows nuclear and cytoplasmic fractions of RelA, the key component of the NF-κB dimer). NF-κB nuclear activity leads to the synthesis of IκBα and A20 (Fig. 1b, d), which constitute the negative feedback loops terminating NF-κB signalling 1.5 h post LPS stimulation (Fig. 1c).

Poly(I:C) (delivered intracellularly by lipofectamine transfection) binds cytosolic RIG-I and MDA5 receptors[24,25], an event that leads to a delayed but longer-lasting activity of NF-κB and IRF3 (Fig. 1b, c). In some cell lines poly(I:C) activates innate immune signalling through the TLR3 receptor, located primarily in the endosomes[26]. We found, however, that in the case when poly(I:C) is delivered by lipofectamine transfection, WT and $Tlr3^{-/-}$ MEFs exhibit similar responses (see Methods). Both RIG-I and MDA5 receptors utilise the same adaptor, mitochondrial antiviral-signalling protein (MAVS)[25], but differ in their ability to bind RNA of varying lengths[27]. Based on the length estimation of poly(I:C) used in our studies (see Methods) we expect that RIG-I is the key receptor responsible for poly(I:C) recognition in our experimental conditions. The resulting *Nfkbia* (IκBα) and *Tnfaip3* (A20) mRNA expression peaks at 4–6 h post-stimulation, much later than for LPS stimulation (Fig. 1d). The mRNA time profiles in response to poly(I:C) resemble those obtained for LPS with cycloheximide (CHX) incubation, which leads to prolonged activity of NF-κB (see further below); however, the mRNA levels in the last case are higher. This suggests that poly(I:C) decreases *Nfkbia* and *Tnfaip3* mRNA stability and inhibits their translation. Stimulation with poly(I:C) and, to a smaller extent, with LPS, activates c-Jun (Fig. 1c), a subunit of AP-1, the transcription factor considered the third component of the IFNβ enhanceosome (in addition to IRF3 and NF-κB)[28].

Fixed-cell staining indicates that NF-κB and IRF3 responses to poly(I:C) are predominantly binary with few cells exhibiting partial translocation (Fig. 2a). All-or-nothing NF-κB and IRF3 responses are also observed for lower poly(I:C) doses, only the fraction of activated cells is lower (Supplementary Fig. 1a, b). Activation of IRF3 and NF-κB is followed by a more widespread activation of STAT1, suggesting autocrine and paracrine activation via IFNβ. At 4–10 h after poly(I:C) stimulation, nuclear NF-κB and IRF3 translocation occurred in about 40% and 25% of cells, respectively (Fig. 2b). Most cells exhibiting IRF3 nuclear translocation also exhibit NF-κB translocation (Fig. 2a, b)—they will be referred to as active cells. The high correlation of IRF3 and NF-κB activation suggests that the 'respond-or-not' decision is made before the IRF3 and NF-κB pathways diverge, and/or that

activation of one pathway enhances activation of the other. This is in line with the observation that IRF3 activation in response to poly(I:C) is weaker and shorter in $RelA^{-/-}$ than wild-type (WT) MEFs (Supplementary Fig. 2). Immunostaining of WT MEFs and time-lapse imaging of RelA-GFP MEFs confirm well-synchronised pulse-like responses to LPS (Fig. 2d, e), previously observed for another MEF cell line (NIH/3T3 RelA-dsRed)[7,8]. In contrast, the responses to poly(I:C) are very heterogeneous, but NF-κB activation is typically switch-like, i.e. once NF-κB enters the nucleus, it remains nuclear for several hours (Fig. 2c). From the high correlation between NF-κB and IRF3 translocations observed for fixed cells we expect that IRF3 activation exhibits similar kinetics.

IFNβ–JAK/STAT signalling stabilises the NF-κB/IRF3 activity: Poly(I:C) stimulation triggers IFNβ synthesis. *Ifnb1* (IFNβ) mRNA level peaks at 4–6 h (Fig. 3a), which coincides with maximum IRF3 phosphorylation (Fig. 1c). Noticeable concentrations of IFNβ are detected in the medium after 6 h and continue rising until 24 h (Fig. 3b). The accumulation of IFNβ is delayed for lower poly(I:C) doses, however, the concentration at 24 h was largely independent of the dose (from 0.1 to 3 μg/ml). This may result from the fact that IFNβ triggers *Ifnb1* transcription at later times (Fig. 3a). Neither high-dose LPS nor TNFα induce IFNβ secretion (Fig. 3b). RelA-deficient MEFs in response to poly(I:C) showed a downregulated *Ifnb1* transcription and secretion when compared to WT cells (Fig. 3a, b). These data add to the evidence that coactivation of NF-κB and IRF3 is required for full activation of *Ifnb1* transcription and secretion. STAT1-deficient cells have a somewhat lowered *Ifnb1* mRNA levels and markedly reduced IFNβ secretion (Fig. 3a, b). The effect of STAT1 knockout was mimicked by using IFNα/β receptor-blocking antibodies (α-IFNAR), administered to WT fibroblasts together with poly(I:C). This suggests that IFNβ secreted by poly(I:C)-stimulated cells acts indirectly via the JAK/STAT pathway to stimulate its own production.

Poly(I:C) stimulation leads to switch-like activation of STAT1 starting at 3–4 h (Figs. 2a and 4a and Supplementary Fig. 1c), i.e. 1–2 h after IRF3 and NF-κB activation (Fig. 1b). STAT1 activation is almost absent after inhibiting IFNAR (Fig. 4a), however, surprisingly, it starts before a substantial accumulation of IFNβ (Fig. 3b). We therefore expect that the secreted IFNβ initially binds IFNAR and activates STATs in the same cell, or its neighbours, and accumulates in the medium only after saturating cell receptors. Poly(I:C)-induced STAT1 activation triggers expression of its target genes, including *Stat1*, *Stat2*, *Ddx58* (RIG-I), *Socs1*, *Eif2ak2* (PKR) and *Oas1a*, which peak at 6–10 h (Fig. 4b, d), and is followed by the accumulation of corresponding proteins (Fig. 4a, c). As expected, the direct IFNβ stimulation results in similar mRNA profiles of STAT-responsive genes, but transcription starts about 2–3 h earlier (Fig. 4b, d). Expression of STAT target genes remains at basal levels after LPS stimulation (Fig. 4b, d), a condition when only NF-κB is activated, and is substantially decreased in RelA-deficient cells in response to poly(I:C) stimulation, a condition when only IRF3 is activated. STAT1 knockout limits activation of all STAT target genes (Fig. 4b, d). The consequent drop of the residual RIG-I level inhibits signal transmission. Consequently, phosphorylation of TBK1 and IKKα/β is suppressed and IRF3 activation is significantly decreased (Fig. 4a). α-IFNAR dampens STAT1 activation after poly(I:C), but the presence of residual RIG-I allows for poly(I:C) recognition, phosphorylation of IKKα/β and TBK1, and subsequent activation of IRF3 (Fig. 4a). Expression of the STAT-dependent genes is decreased in case of α-IFNAR treatment but to a lesser extent than in STAT1-deficient cells (Fig. 4b, d).

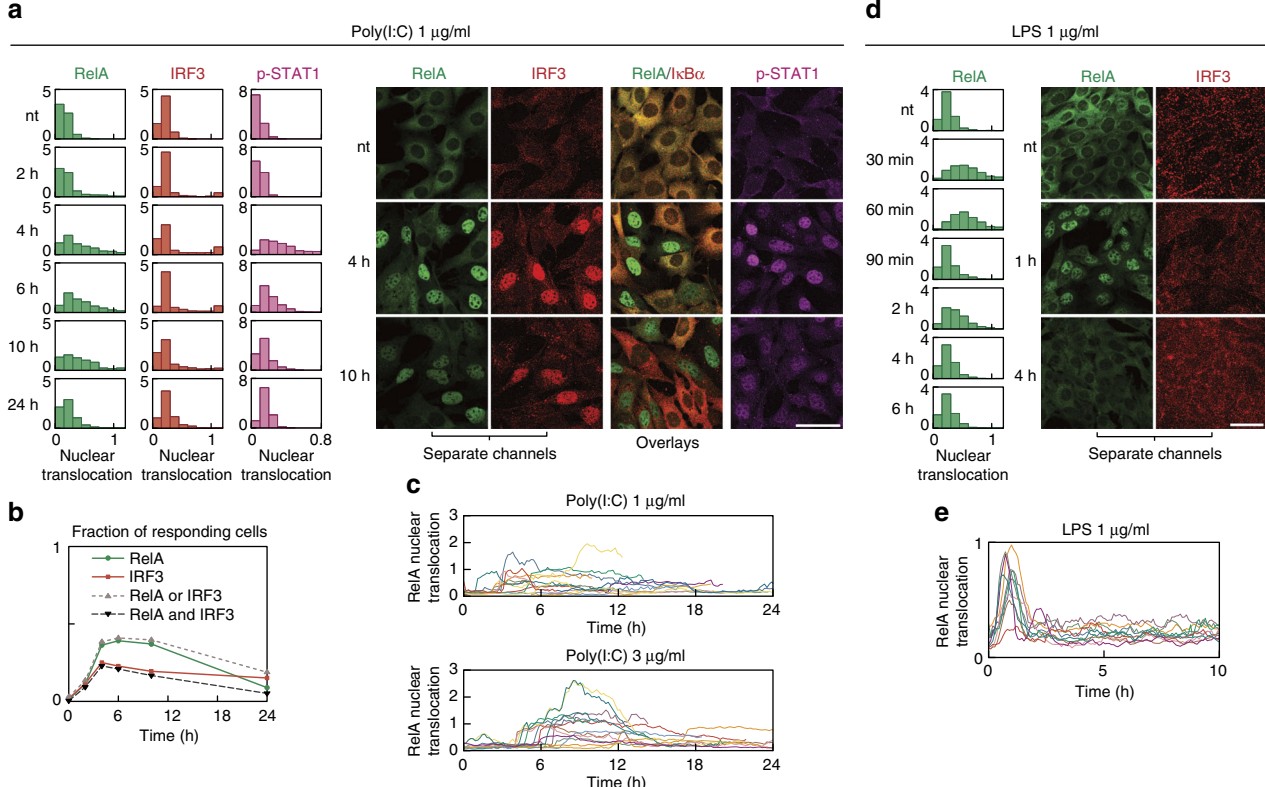

**Fig. 2** Nuclear dynamics of NF-κB and IRF3: pulse-like after LPS, switch-like binary after poly(I:C). Nuclear translocation of a transcription factor is defined here throughout as a normalised quantification of its nuclear fluorescence in confocal images. RelA and IRF3 use a normalisation relating nuclear to whole-cell fluorescence, which underlies the histograms, fractions of responding cells and scatter plots. Distinct normalisations were used for p-STAT1 (to account for its changing cellular level) and for trajectories in RelA-GFP MEFs (see Methods). **a, b** Time course of nuclear localisation of RelA (NF-κB), IRF3, IκBα and p-STAT1 in response to poly(I:C). Scale bars: 50 μm. WT MEFs were stimulated with 1 μg/ml poly(I:C), fixed and stained at given time-points with antibodies against RelA and IRF3, RelA and IκBα, or p-STAT1. **a** Histograms show the full time course. Representative excerpts from confocal images show cells after 0 (nt), 4 and 10 h of poly(I:C) stimulation. **b** Changes in fraction of cells responding to poly(I:C) in time. A cell was deemed responding if its nuclear translocation exceeded a threshold based on nuclear fluorescence of non-treated cells. **c** Nuclear RelA trajectories in RelA-GFP MEFs stimulated with 1 μg/ml or 3 μg/ml poly(I:C) for 24 h. Sample time-lapse confocal microscopy images showing nuclear translocation of GFP-tagged RelA in response to poly(I:C) are provided in Supplementary Movies 1 and 2. **d** Time course of nuclear localisation of RelA and IRF3 in response to LPS. WT MEFs were stimulated with 1 μg/ml LPS, fixed and stained at given time-points with antibodies against RelA and IRF3. Histograms show the full time course. Representative confocal images show cells after 0 (nt), 1 and 4 h of LPS stimulation. **e** Nuclear RelA trajectories in RelA-GFP MEFs stimulated with 1 μg/ml LPS for 10 h. Sample time-lapse confocal microscopy images showing nuclear translocation of GFP-tagged RelA in response to poly(I:C) are provided in Supplementary Movie 3. Histograms ($n \geq 500$) show a representative experiment out of 3. See Supplementary Data 1 and 2 for corresponding uncropped immunostaining images

Among STAT1/2 targets, PKR and OAS1A are of special interest, as they are responsible for inhibiting translation and for mRNA degradation, respectively. Importantly, both proteins are activated by poly(I:C)[29,30]; PKR is activated by phosphorylation about 2 h after poly(I:C) stimulation (Fig. 4c). We propose that in response to poly(I:C) stimulation, PKR and OAS1A reduce synthesis of NF-κB-inducible inhibitors, IκBα and A20, leading to stabilisation of NF-κB and IRF3 signalling. To explore the concept that the differences between NF-κB activation profiles in response to LPS and poly(I:C) depend on inhibition of translation, we compared responses of MEFs to LPS in the presence or absence of CHX. When translation is inhibited by CHX costimulation, IκBα is not resynthesized and consequently in nearly all cells NF-κB remains in the nucleus for over 4 h after LPS stimulation (Fig. 5a). This resembles the switch-like dynamics observed in response to poly(I:C) in a fraction of cells (Fig. 2a, d). Next, we show that treatment with PKR inhibitor C16, partially blocking PKR, results in the increase of IκBα and A20 levels at 2–4 h after poly(I:C) stimulation, compared to DMSO control (Fig. 5b). This reduces phosphorylation and

nuclear translocation of NF-κB and IRF3 (Fig. 5c) as well as active cell fraction 2–4 h after stimulation (Fig. 5d).

Concluding, in response to poly(I:C), secreted IFNβ activates the JAK/STAT pathway, which leads to upregulation of STAT1/2, RIG-I, OAS1A and PKR. These proteins mediate a multi-layer positive-feedback coupling the NF-κB and IRF3 pathways with the JAK/STAT pathway. This positive feedback is effective in response to poly(I:C) as it leads to simultaneous induction of NF-κB and IRF3, and posttranslational activation of OAS1A and PKR, which suppress synthesis of inhibitors, IκBα and A20. In the previous study on A549 cells[31], focused on early stage of response to dsRNA, the paracrine mediated positive feedback was not analysed. Recently, Ourthiague et al. (2015)[32] analysed consequences of another positive feedback, in which ISGF3 (STAT1/STAT2/IRF9 complex) regulates expression of IFNβ. This paracrine feedback could potentially lead to an uncontrolled IFNβ expression in cell population. The IFNβ storm could be avoided if ISGF3 complex, which binds the same motif as IRF3, due to its size may be sterically impeded by AP-1 or NF-κB from binding to the IFNβ enhanceosome.

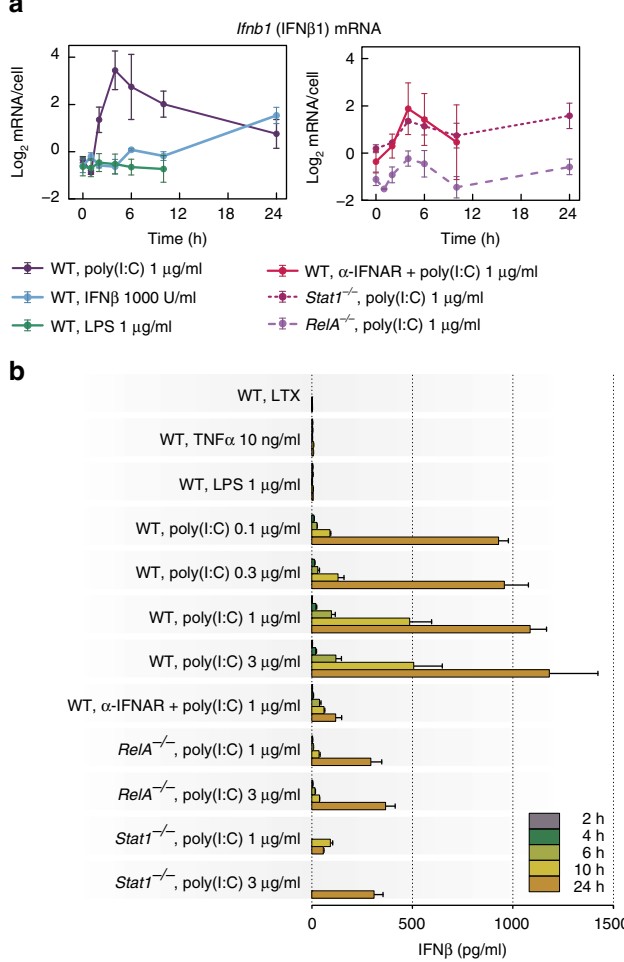

**a**

*Ifnb1* (IFNβ1) mRNA

Log₂ mRNA/cell vs Time (h)

- WT, poly(I:C) 1 µg/ml
- WT, IFNβ 1000 U/ml
- WT, LPS 1 µg/ml
- WT, α-IFNAR + poly(I:C) 1 µg/ml
- *Stat1*⁻ᐟ⁻, poly(I:C) 1 µg/ml
- *RelA*⁻ᐟ⁻, poly(I:C) 1 µg/ml

**b**

- WT, LTX
- WT, TNFα 10 ng/ml
- WT, LPS 1 µg/ml
- WT, poly(I:C) 0.1 µg/ml
- WT, poly(I:C) 0.3 µg/ml
- WT, poly(I:C) 1 µg/ml
- WT, poly(I:C) 3 µg/ml
- WT, α-IFNAR + poly(I:C) 1 µg/ml
- *RelA*⁻ᐟ⁻, poly(I:C) 1 µg/ml
- *RelA*⁻ᐟ⁻, poly(I:C) 3 µg/ml
- *Stat1*⁻ᐟ⁻, poly(I:C) 1 µg/ml
- *Stat1*⁻ᐟ⁻, poly(I:C) 3 µg/ml

IFNβ (pg/ml)

2 h / 4 h / 6 h / 10 h / 24 h

**Fig. 3** Transcriptional activation and secretion of IFNβ. **a** Time profiles of *Ifnb1* (INFβ) mRNA levels in response to different stimuli. WT MEFs were stimulated with 1000 U/ml IFNβ, 1 µg/ml LPS, or 1 µg/ml poly(I:C) in the absence (left) or presence (right) of IFNAR-blocking antibody (α-IFNAR). *Stat1*⁻ᐟ⁻ MEFs and *RelA*⁻ᐟ⁻ MEFs (right) were also stimulated with 1 µg/ml poly(I:C). α-IFNAR was added to cells at 10 µg/ml at 0, 3, 6 and 10 h after poly(I:C) transfection. Time profiles of relative mRNA levels were obtained with RT-PCR, and then rescaled to absolute numbers using digital PCR measurements (bars represent means ± s.e.m., $n \geq 2$; see Supplementary Note for plots of all replicates compared with model simulations). **b** Secretion of IFNβ in response to TNFα, LPS or poly(I:C). WT MEFs were stimulated with 10 ng/ml TNFα ($n = 2$), 1 µg/ml LPS, 0.1, 0.3, 1 and 3 µg/ml poly(I:C), or 1 µg/ml poly(I:C) in the presence of α-IFNAR ($n = 2$). Both *Stat1*⁻ᐟ⁻ MEFs and *RelA*⁻ᐟ⁻ MEFs were stimulated with 1 µg/ml or 3 µg/ml poly(I:C). For each condition, IFNβ concentration after 2, 4, 6, 10 and 24 h was measured by ELISA in 200 µl of culture medium harvested from above 25,000 ± 5000 cells. Bars represent means ± s.e.m. ($n = 3$, except where stated otherwise, values of all replicates are provided in Supplementary Table 8)

Single-cell data show that responses to poly(I:C) are heterogeneous (Fig. 2). There are at least two potential sources of this heterogeneity: uneven partition of poly(I:C) between cells due to lipofectamine-based delivery and initial variability of key proteins levels. Analysis of cellular uptake of fluorescent poly(I:C) (Supplementary Movie 1) indicates a broad distribution of cellular poly(I:C) at 4 h after stimulation, i.e., when most of poly(I:C) enters cells but is not yet degraded (Supplementary Fig. 3). The uneven uptake of poly(I:C) causes that the fraction of

cells exhibiting strong nuclear translocation is smaller. Both experiment and the model indicate that the response variance (at the level of NF-κB and IRF3 activation) is higher than the input variance (at the level of cellular poly(I:C) distribution), which indicates that heterogeneity in initial cell states also contributes to response variability (Supplementary Fig. 4).

**Mathematical model**. The data discussed so far allowed us to uncover the dynamical structure of the system (Fig. 6a) and develop a stochastic model accounting for the observed heterogeneity in cellular responses. We used the model-specification language of BIONETGEN and the associated software[33] that allows for efficient deterministic and stochastic simulation employing the Gillespie algorithm[34]. A full description of the model and a systematic comparison of experimental and simulated mRNA profiles are provided in the Supplementary Note. Computational codes are provided in Supplementary Data 15.

Negative feedbacks via IκBα and A20: NF-κB and IRF3 activity is controlled by negative feedbacks mediated by NF-κB-inducible inhibitors: IκBα and A20. In resting cells, NF-κB is sequestered by IκBα in the cytoplasm. Upon LPS or poly(I:C) stimulation, IκBα is degraded, NF-κB enters the nucleus and triggers transcription of genes coding for IκBα and A20. Resynthesized IκBα enters the nucleus, binds to NF-κB, and directs it back to the cytoplasm. A20 interferes with the upstream NF-κB and IRF3 signalling by inhibiting kinases IKK and TBK1. In the context of LPS stimulation, negative feedback loops mediated by IκBα and A20 lead to pulse-like cytoplasmic-to-nuclear NF-κB shuttling (Fig. 2e and Supplementary Fig. 5a). In our MEFs-specific model, LPS stimulation does not lead to IRF3 activation.

Poly(I:C) stimulation suppresses IκBα and A20 feedbacks: The IκBα- and A20-mediated negative feedbacks are compromised when protein synthesis is suppressed and/or mRNA degradation is enhanced. This happens for LPS and CHX costimulation leading to a stable NF-κB activation (Fig. 5a and Supplementary Fig. 5b). Poly(I:C) activates IKK and TBK1 kinases, leading to the degradation of IκBα and phosphorylation of IRF3, inducing nuclear translocation of NF-κB and IRF3 (Fig. 1c and Supplementary Fig. 6). Poly(I:C) activates OAS1A and PKR post-translationally, inhibiting IκBα and A20 synthesis, and thus allowing for stabilisation of NF-κB and IRF3 activity. This mechanisms is effective only in the fraction of cells having high initial levels of RIG-I, OAS1A and PKR (Fig. 2a and Supplementary Fig. 6).

Positive feedback and feedforward via JAK/STAT: Simultaneous activation of NF-κB and IRF3 triggers synthesis of IFNβ, which activates the JAK/STAT pathway in a paracrine and autocrine manner. Phosphorylated STAT1 and STAT2 dimerise, translocate to the nucleus and trigger transcription of *Ddx58* (RIG-I), *Eif2ak2* (PKR) and *Oas1a*, as well as *Stat1/2*. The resulting increased level of RIG-I further enhances signal transmission, while increased PKR and OAS1A levels enhance inhibition of IκBα and A20 synthesis after poly(I:C) stimulation.

Negative feedback via SOCS1 silences STAT1/2 activity: STAT1/STAT2 heterodimers and STAT2 homodimers serve as transcription factors for *Stat1* and *Stat2*. This can potentially lead to an uncontrolled rise of STATs levels and activity, however this positive-feedback loop is controlled by STAT1/2-inducible SOCS1, which inhibits IFNAR receptors and attenuates STAT1/2 phosphorylation[35,36]. As a result, although total STAT1/2 levels increase for 24 h after poly(I:C) stimulation, STAT1 phosphorylation peaks at 6–10 h (Fig. 4a and Supplementary Fig. 6). The SOCS1-mediated negative feedback allows for long-lasting IFNβ priming. Primed cells remain sensitive for a long time with

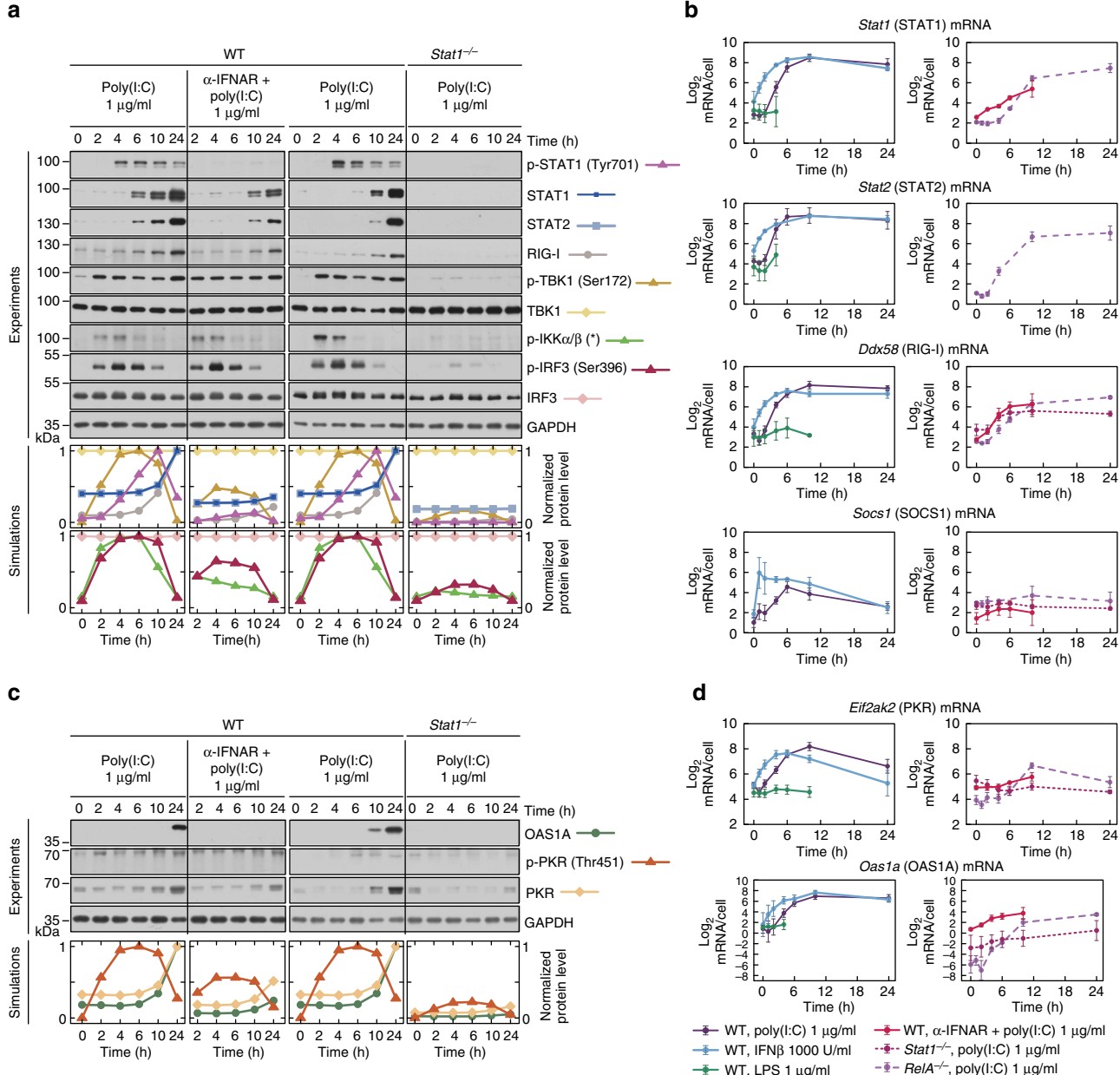

**Fig. 4** Paracrine regulation via the JAK/STAT pathway. **a**, **c** Protein levels of the pathway components in response to 1 µg/ml poly(I:C), characterised by western blotting and numerical model simulations. In separate experiments WT MEFs were stimulated with poly(I:C) in the presence of α-IFNAR, and *Stat1*⁻/⁻ MEFs were stimulated with poly(I:C), for the indicated times. For each experiment, its control, poly(I:C)-stimulated WT MEFs, is included. Whole-cell extracts were analysed. Representative replicates out of 3 are shown. Trajectories show averages of over 200 independent stochastic simulations; the colour key is located next to protein labels. **a** Mediators of positive feedback were analysed using antibodies against phosphorylated (active) forms of STAT1 (p-Tyr701), TBK1, IKKα/β and IRF3 (p-Ser396), as well as total STAT1, STAT2, RIG-I, TBK1, and IRF3. (*) = IKK isoform-dependent phosphorylation sites: p-IKKα Ser176/180, p-IKKβ Ser 177/181. **c** STAT1/2-regulated mediators of double negative feedback were analysed using antibodies against total OAS1A and PKR, as well as for a phosphorylated (active) form of PKR (p-Thr451). **b**, **d** mRNA levels of STAT1/2-regulated genes, *Stat1*, *Stat2*, *Ddx58* (RIG-I), *Socs1*, *Eif2ak2* (PKR) and *Oas1a*, in response to LPS, IFNβ or poly(I:C). WT MEFs were stimulated with 1000 U/ml IFNβ, 1 µg/ml LPS or 1 µg/ml poly(I:C) in the absence (left) or presence (right) of IFNAR-blocking antibody (α-IFNAR). *Stat1*⁻/⁻ MEFs and *RelA*⁻/⁻ MEFs (right) were also stimulated with 1 µg/ml poly(I:C). Time profiles of relative mRNA levels were obtained with RT-PCR, and then rescaled to absolute numbers using digital PCR measurements (bars represent means ± s.e.m., *n* ≥ 2, see Supplementary Note for plots of all replicates compared with model simulations)

increased levels of RIG-I, PKR, OAS1A and STATs, but with low transcription of these components' genes.

Population heterogeneity: Single-cell responses are heterogeneous and only a fraction of cells respond with a coordinated activation of NF-κB and IRF3 to poly(I:C) stimulation. In the model we account for two sources of heterogeneity: uneven partition of poly(I:C) between cells and the uneven initial distribution of RIG-I, PKR and OAS1A[37–40]. Digital PCR data indicate that these components have low basal expression, of the order of 10 mRNA/cell (for *Ddx58* (RIG-I) and *Eif2ak2* (PKR)) and 1 mRNA/cell for *Oas1a* (Fig. 4b, d). This is in line with a low basal expression of *Ifnb1* (of order of 1 mRNA/cell, Fig. 3a) and,

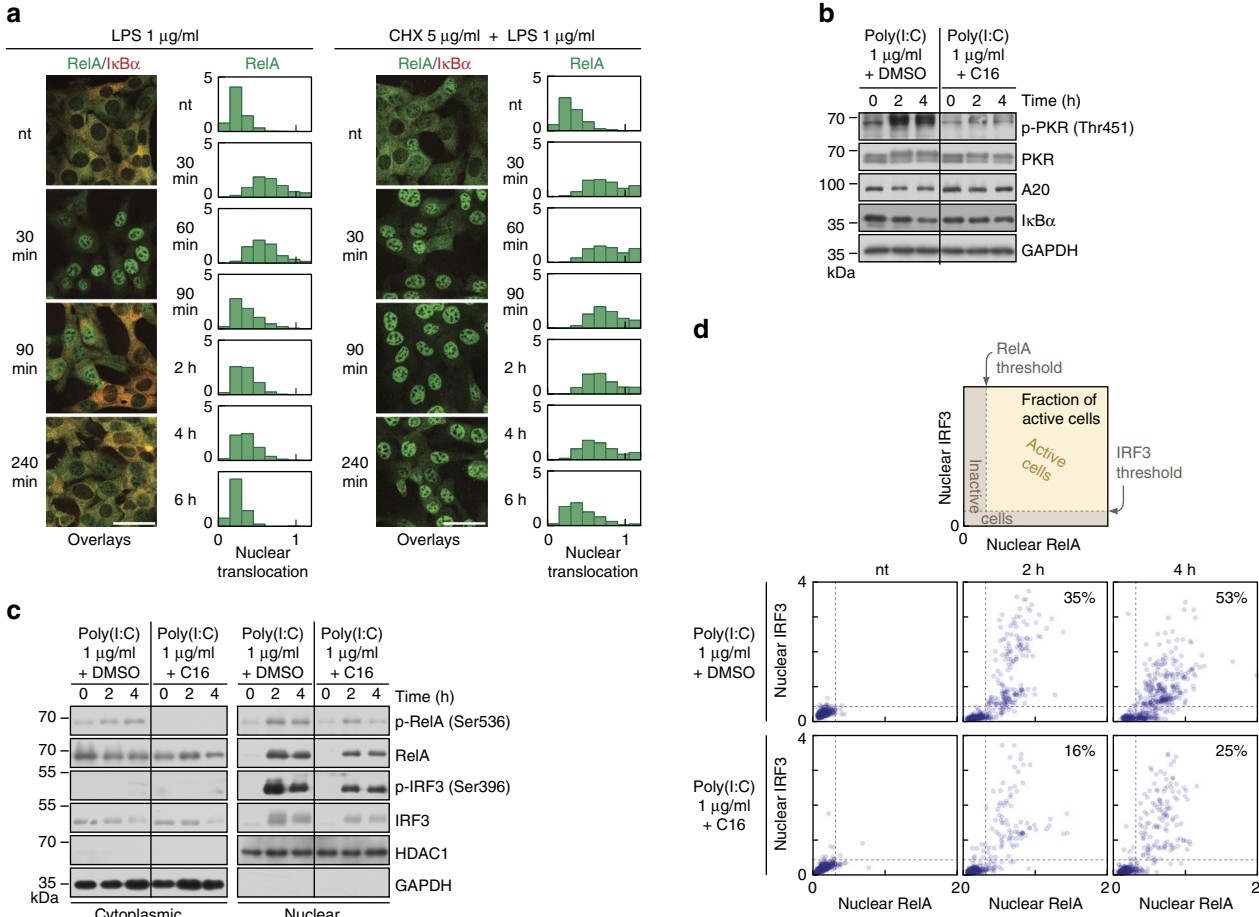

**Fig. 5** Inhibition of translation stabilises translocation of RelA and IRF3. **a** RelA (NF-κB) translocation and cytoplasmic IκBα levels in response to LPS or CHX + LPS. WT MEFs were stimulated with 1 μg/ml LPS in the absence or presence of 5 μg/ml CHX, fixed and stained at given time-points with antibodies for RelA and IκBα. Representative excerpt from confocal images show cells at 0 (nt), 30, 90 and 240 min after LPS stimulation. Histograms ($n \geq 700$, from a representative experiment out of 2) show the full time course of RelA nuclear translocation, defined for Fig. 2. See Supplementary Data 3 for corresponding uncropped immunostaining images. Scale bar: 50 μm. **b–d** Protein levels in response to poly(I:C) upon PKR inhibition. WT MEFs were stimulated with 1 μg/ml poly(I:C) for 0 (nt), 2 and 4 h in the absence or presence of imidazolo-oxindole PKR inhibitor (1 μM/ml), C16, added at 1 h prior to poly(I:C) transfection. Culture medium for all conditions contained the C16 solvent DMSO (0.5% final concentration), and was FBS-free to prevent interference with C16. **b** Whole-cell extracts were analysed using antibodies against total PKR, A20 and IκBα, as well as against a phosphorylated (active) form of PKR (p-Thr451). **c** Nuclear and cytoplasmic fractions were analysed using antibodies against total RelA and IRF3, as well as against their phosphorylated (active) forms, p-RelA (Ser536) and p-IRF3 (Ser396). **d** Cells were fixed and immunostained for RelA and IRF3. Scatter plots show nuclear translocations of RelA vs. IRF3 ($n = 500$) based on confocal images analysis. Percentages indicate fractions of active cells; activity was defined by responding (see also Fig. 2b) with both RelA and IRF3 translocation, as illustrated in a mock plot at the top. See Supplementary Data 4 for corresponding uncropped immunostaining images

correspondingly, low STAT1 basal activity (Fig. 4a). To account for bursty eukaryotic transcription[41,42] we assume that each gene is either ON or OFF, and that gene state switching results from binding and dissociation of transcription factors. Transcriptional bursts during the prestimulation phase lead to a broad distribution of RIG-I and PKR levels at the time of poly(I:C) stimulation, which together with uneven poly(I:C) uptake lead to the response heterogeneity (Supplementary Fig. 4). Randomness causes the deterministic model approximation to produce trajectories different from stochastic trajectories and also different from the stochastic population average (Supplementary Fig. 6, NF-κB$_{nuclear}$).

**IFNβ priming enhances population response and apoptosis.** The stochastic model simulations show heterogeneous activation of NF-κB and IRF3 in response to poly(I:C) and a much more robust activation of these factors in IFNβ-prestimulated cells (Fig. 6b, c). This results from RIG-I accumulation throughout the 24 h long prestimulation phase. In turn, upregulation of IFNAR inhibitor SOCS1 causes that the levels of active STAT1/2 after poly(I:C) are lower in prestimulated cells than in naïve cells (Fig. 6b, c). Accumulation of PKR and OAS1A during the IFNβ-prestimulation phase and their subsequent activation by poly(I:C) reduces levels of IκBα and A20 with respect to naïve cells, allowing for stabilisation of NF-κB and IRF3 activity (Supplementary Fig. 7). To verify these predictions, we compare responses of IFNβ-prestimulated and naïve cells to poly(I:C) (Fig. 7). IFNβ priming activates STAT1 (Fig. 7a, c), increases the levels of positive-feedback components and consequently increases the strength of response to poly(I:C) (Fig. 7a) and fraction of activated cells (Fig. 7b, d, e quantified in Fig. 7f). IFNβ-primed cells exhibit an increased nuclear translocation of NF-κB, accompanied by decreased levels of IκBα (Fig. 7a). The

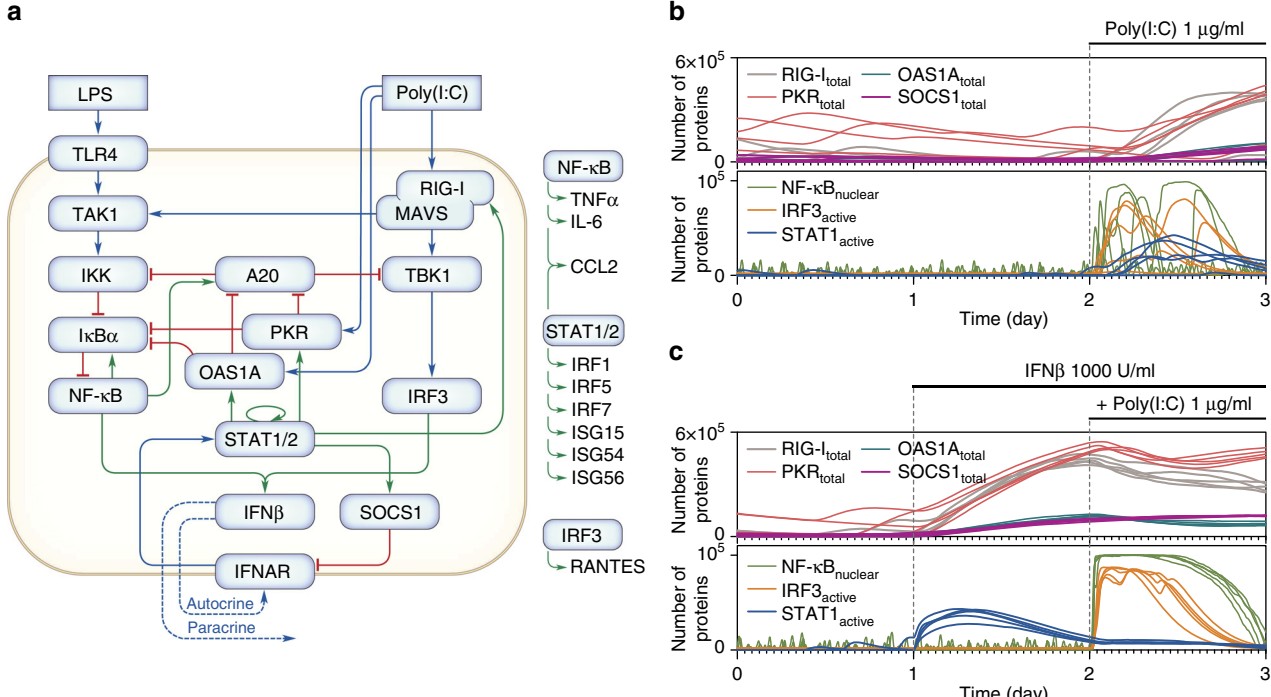

**Fig. 6** Mathematical model: analysis of the effect of IFNβ pre-stimulation on responses to poly(I:C). **a** Network diagram. The components of the model which do not influence other components are listed next to the diagram. Blue arrow-headed lines = activation, red hammer-headed lines = inhibition, green arrow-headed lines = positive regulation of transcription. **b**, **c** The three-day long stochastic simulations are preceded by $3 \times 10^6$ s (~35 days) of stochastic equilibration phase (not shown) rendering different values of model variables at time = 0 for the 5 trajectories shown in each panel. **b** After 2 days of quiescence, WT MEFs were stimulated with 1 µg/ml poly(I:C) for 1 day. **c** After 1 day of quiescence, WT MEFs were prestimulated with 1000 U/ml IFNβ for 1 day, followed by 1 day stimulation with 1 µg/ml poly(I:C)

translocation of IRF3 is even more pronounced, as indicated by a more reddish hue of stained nuclei at 4 h of poly(I:C) stimulation in IFNβ-primed cells (Fig. 7d versus 7b). This confirms that A20, strongly inhibited after the IFNβ prestimulation (Fig. 7a), is a potent inhibitor of the IRF3 pathway[43]. The IFNβ priming has negligible effect for STAT1-deficient cells, which respond weakly to poly(I:C) regardless of prestimulation.

Spread of a viral infection is restricted by three main defence mechanisms. The first involves the upregulation and activation of a set of proteins (such as PKR and OAS1A), which attenuate virus replication. The second, more radical mechanism is the apoptosis of infected cells. The third is based on paracrine IFNβ signalling, which alerts non-infected cells. Poly(I:C) stimulation (in contrast to LPS or TNFα) leads to apoptosis, with the fraction of apoptotic cells increasing in time and reaching about 25% at 24 h (Fig. 8a, e). NF-κB, which in the context of TNFα stimulation plays an anti-apoptotic role, is pro-apoptotic in the context of the poly(I:C) stimulation (Fig. 8b), as it is required for IFNβ synthesis. Although IFNβ itself did not induce cell death, the apoptotic cell fraction increased up to 70–80% in IFNβ-prestimulated cells (Fig. 8c, f). This effect is absent in STAT1-deficient cells, for which apoptotic fraction after poly(I:C) is 15%, regardless of IFNβ-prestimulation (Fig. 8d).

## Discussion

Signalling pathways are sometimes perceived as information transmitting channels, however, the recently demonstrated low information capacity (of order of 1 bit) of NF-κB and MAPK pathways[44,45] suggests that their role is rather to process than to transmit signals. Signal processing relies on regulatory feedback loops. Systems governed by negative feedbacks, exemplified here by the NF-κB system, in response to tonic stimulation (here with

LPS) produce oscillatory or pulsed responses, and can adapt to the constant signal. Positive feedbacks enable switch-like kinetics and can be harnessed for cell fate decision-making. Coupled negative and positive feedbacks give rise to more sophisticated regulatory mechanisms. Importantly, when fast negative feedbacks are suppressed by positive feedbacks acting over longer time scales, the decision process is postponed so that secondary signals can be integrated before a potentially irreversible decision is made[46,47]. We demonstrated that the network underlying the innate immune system has such a topology. Here, negative feedback loops mediated by A20 and IκBα can be antagonised by positive-feedback mediators, OAS1A and PKR, which, when activated by poly(I:C), enhance mRNA degradation and inhibit translation. In the innate immunity network an additional layer of control is introduced by paracrine signalling, which coordinates responses at the cell population level. IFNβ-primed cells upregulate the components of the positive feedback, PKR, OAS1A and RIG-I, which can be rapidly activated upon poly(I:C) stimulation. This shortens the decision phase and facilitates robust commitment to apoptosis, the purpose being to limit the spread of infection.

The constructed mathematical model identifies the role of stochasticity and of the regulatory modules that render the complex network capable of proportionate decision-making at the cell population level. The A20- and IκBα-mediated negative feedbacks terminate LPS- or cytokine-induced signalling. These feedbacks filter out weak signals in the context of poly(I:C) stimulation; as a result, although the cellular responses are 'all-or-nothing', the fraction of responding cells decreases with a decreasing poly(I:C) dose. In response to poly(I:C), positive feedbacks mediated by IFNβ–JAK/STAT axis allow for stabilisation of the activity of NF-κB and IRF3, commitment to the

antiviral state, and apoptosis. Finally, the negative feedback mediated by a STAT inhibitor, SOCS1, allows for the long-lasting IFNβ priming that initially increases the levels of positive-feedback components and subsequently shuts off their transcription. The proposed model explains how the innate immune regulatory network discriminate between bacterial and viral signals, responding by either adaptation or apoptosis.

The analysed mechanisms suggest that in the case of viral infection the first tier of responding cells produce IFNβ and eventually commit to apoptosis. In these cells, due to relatively low initial levels of RIG-I, PKR and OAS1A, the decision phase can last relatively long, allowing the virus to replicate. The second tier consists of cells alerted by IFNβ prior to infection. Increased levels of RIG-I, PKR and OAS1A in these cells allow them to shorten the decision phase and block the spread of virus by

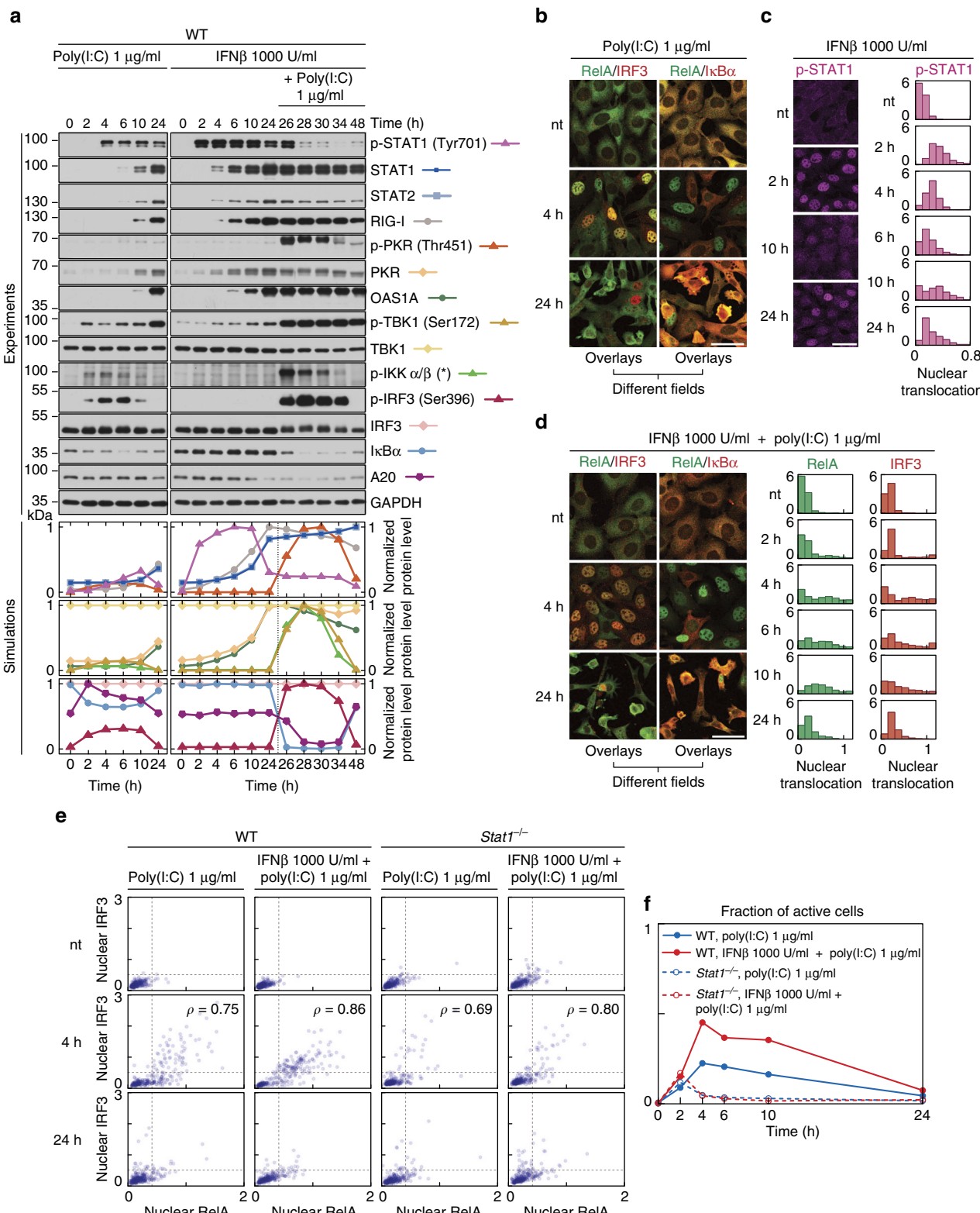

entering the antiviral state and prompt apoptosis. In the third tier, IFNβ-primed cells remain in the alert state due to high levels of slowly degrading STATs. Stochasticity embedded in cell regulation allows the cell population to be guarded only by a small fraction of sensitive cells (expressing high level of RIG-I, PKR and OAS1A). In the emerging scenario, the initial heterogeneity and cooperation of the intracellular negative feedbacks with the positive feedforward paracrine signalling produces cell fate decisions coordinated at the cell population level.

## Methods

**Cell lines and culture.** Experiments were performed on both wild-type (WT) and modified mouse embryonic fibroblasts (MEFs), including MEFs with double knockout of NF-κB subunit RelA (MEF $RelA^{-/-}$), a derivative cell line with stably transfected RelA-GFP reporter construct (MEF RelA-GFP), and MEFs with double knockout of STAT1 (MEF $Stat1^{-/-}$). The original WT cell line and the MEF $RelA^{-/-}$ and MEF RelA-GFP cell lines[48] were developed in the Allan Brasier laboratory. The MEF $Stat1^{-/-}$ cell line[49] was obtained from Prof. Thomas Decker (Max F. Perutz Laboratories, Wien). $Tlr3^{-/-}$ PMEFs, together with the concurrently established WT line, were purchased from OrientalBioService, Inc. Those cells were immortalised using the SV40 T antigen protocol. To this end, cells were transfected with pBABE-neo largeTcDNA plasmid, a gift from Bob Weinberg (Addgene plasmid # 1780)[50]. Immortalised cells were selected through neomycin treatment and serial passaging. Knockout of $Tlr3$ was verified through PCR according to the OrientalBioService protocol (Supplementary Fig. 9a). The following pair of primers were used: 5′-ca gag cct ggg taa gtt att gtg ctg-3′ and 5′-tcc aga caa ttg gca agt tat tcg ccc-3′.

All cell lines were routinely tested against mycoplasma contamination by DAPI staining and PCR. The cells were cultured in adherent conditions on tissue culture-treated dishes or multi-well plates (Falcon) using Dulbecco's modified Eagle medium (DMEM) with 4.5 g/l of D-glucose and 0.1 mM L-glutamine (ThermoFischer Scientific), supplemented with 10% fetal bovine serum (ThermoFischer Scientific), 1% pyruvate (Sigma-Aldrich), 0.1 mM non-essential amino acids (Sigma-Aldrich) and 100 mg/ml penicillin/streptomycin mix (Sigma-Aldrich). Cells were grown and maintained in a conditioned incubator at 37 °C, 5% $CO_2$ and subcultured upon reaching 90% confluency. Cells were counted using TC10 Automated Cell Counter (Bio-Rad). For stimulation, cells were seeded on dishes, multi-well plates or coverslips, depending on the type of experiment and allowed to adhere overnight at 37 °C.

**Compounds and stimulation protocols.** Poly(I:C) was purchased from Sigma-Aldrich, with the mean molecular mass ranging from 200 to 500 kDa, as stated by the manufacturer. Poly(I:C) was delivered to cells by means of lipid-based transfection, using Lipofectamine LTX with Plus Reagent (ThermoFischer Scientific). Manufacturer's protocol was optimised for MEFs transfection. Medium was changed to antibiotic-free DMEM 4 h after seeding, before overnight incubation. Poly(I:C) was mixed with Plus Reagent diluted in serum-free DMEM and then mixed with DMEM-diluted Lipofectamine. Liposome-poly(I:C) complexes were allowed to form for 20 min before adding them to cells. Per $2 \times 10^5$ cells in 1 ml DMEM on a 30 mm dish poly(I:C) was added together with 6 µl of Lipofectamine and 6 µl of Plus Reagent, diluted in 100 µl of serum-free DMEM. These amounts were appropriately scaled up or down for use with different numbers of cells and medium volumes in distinct experiments. Supplementary Movie 1 shows liposome–poly(I:C) fusing with RelA-GFP MEF cells and triggering RelA translocation.

As it is possible that Lipofectamine facilitates poly(I:C) delivery not only to the cytoplasm, where it activates RIG-I (and/or MDA5), but also to endosomes, where it can activate TLR3, it is important to determine the (potential) contribution of TLR3 in innate immune signalling. We thus verified that poly(I:C) (delivered by lipid-based transfection) induces the same activation of transcription factors NF-κB, IRF3, and STAT1 (Supplementary Fig. 9b, c), and leads to the same increase of protein levels of RIG-I, PKR, and OAS1A (Supplementary Fig. 9d) in WT and $Tlr3^{-/-}$ MEFs. The apoptotic rates after poly(I:C) either with or without IFNβ prestimulation were somewhat lower for $Tlr3^{-/-}$ cells (Supplementary Fig. 9e), whereas the fraction of cells showing activation of NF-κB was somewhat lower for WT cells. Moreover, our estimates showed poly(I:C) length to be within the 100–1000 bp range (see Supplementary Data 12), corresponding to the molecular mass range of 67–670 kDa. Poly(I:C) of that length is preferentially bound by RIG-I, while MDA5 is known to bind longer chains[27]. IFNβ secretion was suppressed in $Rig-I^{-/-}$ MEFs in response to short-chain poly(I:C) stimulation[27].

Lipopolysaccharide (LPS) from *Escherichia coli* 0111:B4 (purified by ion-exchange chromatography, Sigma-Aldrich) was stored in 1 mg/ml aliquots. In order to disrupt LPS micelles, it was solubilised in a bath sonicator for 15 min and vortexed vigorously for additional 1 min prior to making further dilutions and adding to cells.

Mouse interferon β (Sigma-Aldrich) was typically used at a concentration of 1000 U/ml. In the prestimulation experiments, it was added to the cells 24 h before poly(I:C) transfection. Cycloheximide (Sigma-Aldrich) was administered to cells at a final concentration of 5 µg/ml, alone or in conjunction with LPS; in the latter case it was added 60 min before LPS.

For interferon receptor blocking experiments, anti-IFNAR blocking antibodies (BD) were added at a concentration of 10 µg/ml simultaneously with poly(I:C) and then supplemented at the same dose after 3, 6 and 10 h in the course of the experiment. Imidazolo-oxindole PKR inhibitor C16 (Sigma-Aldrich) was added to cells at 1 µM concentration 1 h before poly(I:C) transfection. 0.5% DMSO was used as a solvent control in experiments involving C16.

All other reagents and kits are listed in Supplementary Table 6. Supplementary Table 1 contains summary of stimulation protocols used for particular cell lines.

**RNA isolation and reverse transcription.** Cells were seeded on 12-well plates at a density of 100,000 cells/well. Upon completed stimulation, cells were washed with PBS and submitted to isolation of total RNA using PureLink RNA Mini Kit (ThermoFischer Scientific), following manufacturer's instructions. Concentration and quality of isolated RNA was determined by measuring UV absorbance of samples diluted 1:100 in 10 mM Tris-HCl (pH 7.5) at 260 and 280 nm, using Multiskan GO Microplate Spectrophotometer (ThermoFischer Scientific). If not used immediately, RNA was stored for later use at −80 °C. Reverse transcription with random primers was performed from about 2 µg of template RNA using High Capacity cDNA Reverse Transcription Kit (ThermoFischer Scientific). Reaction was performed in Mastercycler Gradient thermal cycler (Eppendorf) under the following conditions: 10 min 25 °C, 120 min 37 °C and 5 min 85 °C.

**Real-time PCR.** RT-PCR was performed on a QuantStudio 12 K Flex Real-Time PCR system with Array Card block (ThermoFischer Scientific). Reverse transcribed cDNA (1000 ng) was mixed with reaction Master Mix and loaded onto TaqMan Array Card containing probes and primers for 16 (in three replicates) or 24 (in two replicates) genes, with endogenous reference controls. Reaction was conducted using QuantStudio 'Standard' protocol, with FAM/ROX chemistry. Upon completion, expression of target genes was analysed using comparative $\Delta C_T$ method with QuantStudio 12K Flex software and normalised against GAPDH gene expression. The $C_T$ values for GAPDH were relatively stable across experiments with various protocols, with the average equal 17.6 and s.d. 0.8 (based on 183 measurements), see Supplementary Data 13. All expression assays used are listed in Supplementary Table 3.

**Digital PCR (dPCR).** Digital PCR measurements were performed using Quant-Studio 3D system (Life Technologies) in order to convert $\Delta C_T$ profiles to absolute numbers of mRNA/cell. In dPCR performed (Supplementary Table 2) we measured the absolute mRNA/cell numbers using the same isolated RNA samples as used for RT-PCR measurements. cDNA reverse transcribed from these samples was diluted to fit into dPCR system detection range and mixed with appropriate gene expression assay and Master Mix (TaqMan, see Supplementary Table 3).

**Fig. 7** Model validation: IFNβ pre-stimulation increases the fraction of responding cells and response strength. Following 24 h of quiescence (**a**, **b**) or prestimulation with 1000 U/ml IFNβ (**a**, **d**), WT MEFs were stimulated with 1 µg/ml poly(I:C) for 0–24 h, or they were only stimulated with 1000 U/ml IFNβ for 0–24 h (**c**). **a** Protein levels of model components, characterised by western blotting and numerical model simulations. Whole-cell extracts were analysed with antibodies against NF-κB–IRF3–STAT1/2 pathways components. (*) = IKK isoform-dependent phosphorylation sites: p-IKKα Ser176/180, p-IKKβ Ser 177/181. Representative experiments out of 2 are shown. Trajectories show averages of over 200 independent stochastic simulations; the colour key is located next to protein labels. **b–d** Cells were fixed and immunostained for (**b**, **d**) RelA (NF-κB) and IRF3 or RelA and IκBα, or (**c**) p-STAT (Tyr701). Representative excerpts from confocal images show cells at the indicated times after stimulation. **c**, **d** Histograms ($n \geq 600$, from one out of 2 experiments) show the full time course of translocation, as defined for Fig. 2. Scale bars: 50 µm. **e**, **f** Following 24 h of quiescence or pre-stimulation with 1000 U/ml IFNβ, WT and $Stat1^{-/-}$ MEFs were stimulated with 1 µg/ml poly(I:C) for 0–24 h, fixed and stained with antibodies against RelA and IRF3. **e** Scatter plots show nuclear translocations of RelA vs. IRF3 (as defined for Fig. 2b, $n = 300$); $\rho$ is the Pearson correlation coefficient. **f** Fractions of active cells (see Fig. 5d) calculated from the scatter plots in **e**. See Supplementary Data 5, 6 and 7, 8 for uncropped immunostaining images corresponding to **c**, **d** and scatter plots for $Stat1^{-/-}$ MEFs in **e**

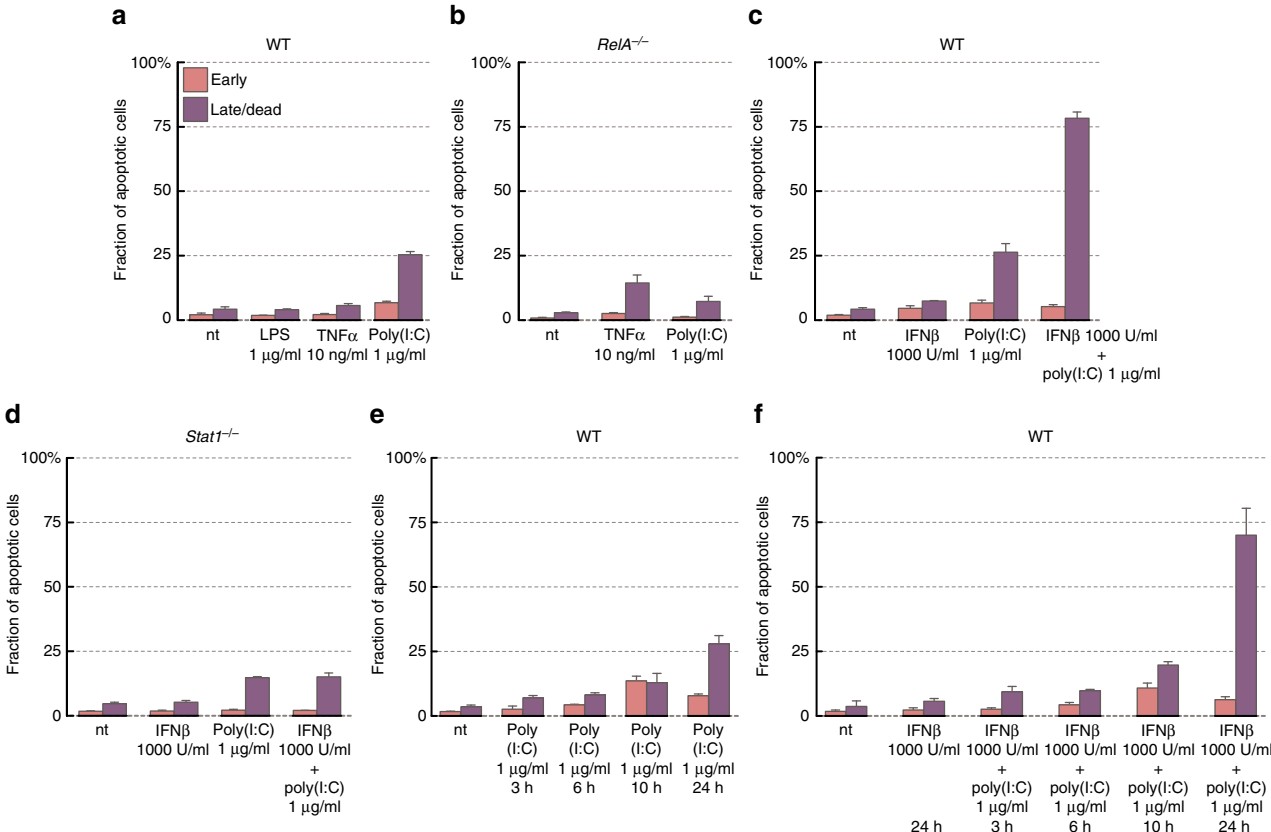

**Fig. 8** Apoptotic cell fraction in response to LPS, TNFα, poly(I:C) or IFNβ. Bars show mean fractions of early apoptotic (pink) and late apoptotic/dead cells (purple), estimated by Annexin V/PI staining (bars represent means ± s.d., $n \geq 3$, data for all replicates are given in the Supplementary Data 14). See Supplementary Fig. 8 for density plots of Annexin V/PI staining. **a** Fraction of apoptotic WT MEFs in response to 1 μg/ml LPS, 10 ng/ml TNFα or 1 μg/ml poly(I:C) after 24 h of treatment. **b** Fraction of apoptotic $RelA^{-/-}$ MEFs in response to 24-hr stimulation with 10 ng/ml TNFα or 1 μg/ml poly(I:C). **c** Fraction of apoptotic WT MEFs in response to 24-hr treatment with 1 μg/ml poly(I:C) following 24 h of quiescence or 1000 U/ml IFNβ pre-stimulation. **d** Fraction of apoptotic $Stat1^{-/-}$ MEFs in response to 24 h of 1000 U/ml IFNβ stimulation, 1 μg/ml poly(I:C) 24 h treatment, or response to 24 h treatment with 1 μg/ml poly(I:C) after 24 h of 1000 U/ml IFNβ pre-stimulation. **e** Apoptotic fraction time course for WT MEFs in response to 1 μg/ml poly(I:C). **f** Apoptotic fraction time course for WT MEFs in response to 1 μg/ml poly(I:C) following 24 h of 1000 U/ml IFNβ pre-stimulation

Sample loaded onto QuantStudio 3D Digital PCR Chip was thermocycled using ProFlex PCR System (ThermoFischer Scientific) according to the manufacturer's instruction. Chips were analysed using QuantStudio 3D Digital PCR Instrument and ANALYSIS SUITE cloud software. The dPCR measurements were used to rescale RT-PCR data to absolute numbers of mRNA/cell:

Based on the simultaneous dPCR and RT-PCR measurements for $i$-th gene, we calculated the normalisation coefficient

$$Q_i = \langle \log_2(\text{mRNA})_i + \Delta C_T \rangle$$

by averaging over $n \geq 3$ dPCR and RT-PCR pairs of measurements performed for different lysates. Coefficients $Q_i$ were then used to convert all RT-PCR data to absolute numbers of mRNA/cell according to the formula

$$\left(\log_2(\text{mRNA})_i\right)_k = Q_i - \left(\Delta C_T(i)\right)_k$$

where $k$ refers to a time-point in a given experiment for $i$-th gene, see Supplementary Table 7. In Supplementary Note, the same relation is used in reverse to calculate $\Delta C_T$ based on an average number of mRNA/cell obtained from numerical simulations (Supplementary Note Figures A–D). The estimates for the genes that code for the proteins that do not regulate other proteins in the pathway, and for which dPCR data were not obtained, are based on the average $Q_i$ value for genes with the absolute quantification.

**ELISA.** For measuring cytokine production, cells were seeded on 96-well plates at a density of 20,000 cells/well. Upon desired stimulation time according to a chosen protocol, culture medium from cells was collected and stored at −20 °C for further analysis. IFNβ levels were estimated using VeriKine Mouse IFNβ ELISA kit (PBL Assay Science). Standards and samples were measured in duplicates or triplicates. Optical densities of samples after final colour development were determined using Multiskan GO Microplate Spectrophotometer (ThermoFischer Scientific) at a wavelength of 450 nm. Cytokine concentrations were obtained by fitting

parameters of the 'One site—total binding' equation to an 8-point standard curve using GRAPHPAD PRISM software.

**Apoptosis level estimation.** Apoptosis was measured using Annexin V and propidium iodide staining method (FITC Annexin V Apoptosis Detection Kit II, BD Bioscience), measuring the amount of phosphatidylserine externalised to the outer layer of cell membrane. For the analysis, cells were seeded on 12-well plates at a density of 100,000 cells/well and treated according to the chosen protocol. Analysis was performed on both the cells attached to the plate surface and dying cells suspended in the medium. Flow cytometry measurement was performed using FACSCalibur (BD) and the CELLQUESTPRO analysis software. Cells which are both Annexin V- and PI-negative were considered viable, Annexin V-positive and PI-negative were considered early apoptotic; PI-positive cells, irrespective of the outcome of Annexin V staining, were considered late apoptotic or dead.

**Immunostaining.** Cells were seeded on 12 mm-diameter round glass coverslips, which were previously washed in 60% ethanol/40% HCl, thoroughly rinsed with water and autoclaved. Seeding density was 50,000 cells/coverslip. After stimulation, cells on coverslips were washed with PBS and immediately fixed with 4% formaldehyde (20 min, room temperature). Cells were then washed thoroughly and incubated for 10 min with 50 mM NH$_4$Cl. Cell membranes were permeabilized with 0.1% Triton X-100 (Sigma-Aldrich) for 5 min, washed again and blocked with 5% BSA/PBS. Antibodies detecting target proteins were then added to the cells in 5% BSA, and incubated for 1.5 h (see Supplementary Table 4). After washing cells 5 times with PBS, appropriate secondary antibodies conjugated with fluorescent dyes were added and incubated for another 1.5 h (Supplementary Table 5). Subsequently, cells were washed and their nuclei were stained for 10 min with 200 ng/ml DAPI (Sigma-Aldrich). Coverslips with stained cells were mounted on microscope slides with a drop of Mowiol (Sigma-Aldrich) or Vectashield (Vector) and observed using Leica TCS SP5 X confocal microscope with LEICA APPLICATION SUITE AF software. For staining cells with anti-phospho-STAT1 antibody (Cell Signaling

Technology, Inc.; see Supplementary Table 4), cells were first fixed with 4% formaldehyde as described above and then permeabilized with ice-cold 100% methanol (20 min, room temperature). After thorough washing with PBS, cells were blocked with 5% BSA/PBS containing 0.3% Triton X-100 (1 h). Anti-phospho-STAT1 antibody was added to cells in the same blocking buffer and incubated overnight. Further procedures were as described above.

**Live-cell imaging**. MEF RelA-GFP cells were seeded on a four-well chambered slide (Lab Tek) at a density of 70,000 cells/well. After letting the cells adhere to the surface for 3–4 h, they were stained for 10 min with 1 µg/ml Hoechst 33342 (Sigma-Aldrich; blue fluorescence-emitting dsDNA stain), washed twice with fresh medium, and left overnight. After placing the slide inside the Leica TCS SP5 X confocal microscope's environmental chamber with controlled atmosphere (37 °C, 5% CO$_2$), stimulation was performed according to the chosen protocol. Observations in real time were performed on at least three fields from each chamber, for up to 24 h as specified, every ~10 min.

**Microscopic image analysis**. Nuclei detection and tracking in confocal images were performed automatically and corrected manually within our in-house software ('MEFTRACK'). For immunostaining images the automatic detection of nuclei was corrected manually by excluding unfit (mitotic, overlapping or otherwise misshapen) nuclei and occasionally re-drawing nuclear contours. In each frame several background regions were marked and their fluorescence was quantified alongside nuclear fluorescence, in all available channels. For time lapse images, automatic nuclei detection in all frames was coupled with exclusion of oversize ('non-splittable') nuclei and debris. All nuclei were tracked automatically based on parameters such as frame-to-frame proximity, surface area, eccentricity, orientation, total fluorescence intensity and intensity distribution (both in the nuclear channel). Nuclei of interest were then individually checked for track consistency and contour fit, both of which were corrected manually where necessary. To quantify the magnitude of nuclear translocation, auxiliary MATLAB scripts were used to quantify nuclear fluorescence intensities.

Nuclear translocation in nucleus $n_i$ is calculated as the ratio of [background-corrected fluorescence intensity sum of all pixels in the $n_i$'s nuclear contour in the primary quantified channel, denoted $\tilde{I}^Q_{n_i}$] to [the background-corrected fluorescence intensity sum of all pixels in the $n_i$'s nuclear contour in the auxiliary nuclear staining quantified channel, denoted $\tilde{I}^N_{n_i}$]. This ratio is normalised using the relative overall image intensity in the primary channel, $\tilde{I}^Q_*$, vs. in the auxiliary channel, $\tilde{I}^N_*$. Taken together,

$$\text{NuclearTranslocation}(n_i) = \frac{\tilde{I}^Q_{n_i}/\tilde{I}^Q_*}{\tilde{I}^N_{n_i}/\tilde{I}^N_*} \qquad (1)$$

Background correction in the primary quantified channel is performed by subtracting the average background pixel intensity in the primary quantified channel, $\langle p^Q_{\text{bg}}\rangle$, multiplied by the $n_i$'s nuclear contour surface area, $S^N_{n_i}$:

$$\tilde{I}^Q_{n_i} = I^Q_{n_i} - S^N_{n_i} \times \langle p^Q_{\text{bg}}\rangle$$

Analogous formulae are used for the auxiliary nuclear staining channel, i.e., background-corrected $\tilde{I}^N_{n_i}$ is obtained from raw $I^N_{n_i}$ according to:

$$\tilde{I}^N_{n_i} = I^N_{n_i} - S^N_{n_i} \times \langle p^N_{\text{bg}}\rangle$$

and for image intensities in the primary and auxiliary nuclear staining channels:

$$\tilde{I}^Q_* = I^Q_* - S^N_* \times \langle p^Q_{\text{bg}}\rangle$$

$$\tilde{I}^N_* = I^N_* - S^N_* \times \langle p^N_{\text{bg}}\rangle$$

Depending on the case, the general formula in Eq. (1) is used with or without modifications:

1. For RelA and IRF3: Eq. (1) is used directly:

$$\text{NuclearTranslocation}_{\text{Case1}}(n_i) = \frac{\tilde{I}^Q_{n_i}/\tilde{I}^Q_*}{\tilde{I}^N_{n_i}/\tilde{I}^N_*}$$

i.e., as described before, background intensity is subtracted in both channels and for each nucleus the ratio of its intensity in the primary channel of interest to its intensity in the nuclear staining channel is divided by the ratio of total image intensity in the nuclear staining channel to total image intensity in the primary channel of interest.

2. For p-STAT1: background intensities are subtracted as in Eq. (1) but total image intensity is not used in normalisation (since the abundance of p-STAT1 in

cells increases in the course of imaged experiments):

$$\text{NuclearTranslocation}_{\text{Case2}}(n_i) = \frac{\tilde{I}^Q_{n_i}}{\tilde{I}^N_{n_i}}$$

3. Live-cell time-lapse images are corrected neither for background intensity nor for total image intensity (since only relative intensity variations in time are of interest):

$$\text{NuclearTranslocation}_{\text{Case3}}(n_i) = \frac{I^Q_{n_i}}{I^N_{n_i}}$$

**Western blotting**. Whole-cell lysates. Cells were seeded on 30 mm tissue culture-treated dishes, at a density of 200,000/dish, and incubated overnight. After stimulation dishes were placed on ice and cells were washed with cold PBS. Cells were harvested by adding 300 µl of modified RIPA buffer (50 mM Tris-HCl pH 7.5, 150 mM NaCl, 1 mM EDTA, 1 mM EGTA, 0.25% sodium deoxycholate, 1% IGEPAL CA-630) supplemented by cocktail of protease and phosphatase inhibitors (cOmplete inhibitor cocktail, Roche; 10 mM sodium fluoride and 1 mM sodium orthovanadate, Sigma-Aldrich). The cell suspension was then transferred into a pre-cooled microcentrifuge tube and allowed to lyse for 30 min on ice. Lysates were centrifuged at 4 °C, 20,000 × g, 20 min, clear supernatant was transferred to a fresh tube and pellet was discarded.

Cell-fractionation: Cells were seeded on a 100 mm tissue culture-treated dishes, at a density of 1,000,000/dish, and incubated overnight. After stimulation, cells were placed on ice, washed with ice-cold PBS, scraped from the dish in PBS and centrifuged (4 °C, 100 × g, 5 min). Cell pellet was then suspended in 1.5 ml of hypotonic cytoplasmic fraction buffer (20 mM HEPES pH 8.0, 0.2% IGEPAL CA-630, 1 mM EDTA, 1 mM DTT, protease and phosphatase inhibitor cocktail, as above) and incubated on ice for 10 min with occasional shaking. After subsequent centrifugation (4 °C, 1700 × g, 5 min), supernatant was set aside and treated as the cytoplasmic fraction; pellet was washed in the same buffer and recentrifuged (as above), and supernatant was discarded. Remaining pellet was suspended in 150 µl of nuclear fraction buffer (20 mM HEPES pH 8, 420 mM NaCl, 20% glycerol, 1 mM EDTA, 1 mM DTT, protein and phosphatase inhibitors, as above), incubated on ice for 30 min with occasional mixing and then centrifuged at 4 °C, 10,000 × g, 10 min. Supernatant containing nuclear fraction was transferred to a fresh tube and left for further processing.

SDS-PAGE and western blot: Part of cell lysate (either whole-cell or fractionated) was diluted 1:100–1:1000 and used to determine protein concentration using Bradford method against a BSA standard. Cell lysate was then subjected to precipitation by adding trichloroacetic acid (TCA) to a final concentration of 10% and keeping on ice for 30 min. Then, lysate was centrifuged at 4 °C, 12,000 × g, 10 min. Supernatant was carefully removed and the pellet was dried by inverting the tube on a tissue paper. Next, protein pellet was washed by adding cold acetone, vortexing and recentrifuging. After centrifugation, acetone was decanted and tube was left open to allow residual acetone to evaporate. Finally, proteins were resuspended in standard Laemmli sample buffer containing 10 mM DTT, and boiled at 95 °C for 10 min. Equal amounts of each protein sample was loaded onto 10% polyacrylamide gel (around 10–50 µg as determined by the Bradford method, depending on the experiment) and denaturing polyacrylamide gel electrophoresis (PAGE) was performed with Mini-PROTEAN Tetra System (Bio-Rad). PageRuler Plus Prestained Protein Ladder (ThermoFischer Scientific) was loaded onto the gel alongside samples.

Upon completion of electrophoresis, proteins were transferred to nitrocellulose membrane using wet electrotransfer in the Mini-PROTEAN apparatus, according to the modified Towbin method (400 mA, 50 min). The membrane was rinsed with TBST (TBS buffer containing 0.1% Tween-20, Sigma-Aldrich) and blocked for 1 h with 5% BSA/TBS or 5% non-fat dry milk. Membranes were incubated at 4 °C overnight with one of the primary antibodies (listed together with dilutions in Supplementary Table 4). After thorough washing with TBST, membranes were incubated with secondary antibodies (listed together with dilutions in Supplementary Table 5) conjugated with horseradish peroxidase for 1 h, at room temperature. After washing, chemiluminescent reaction was developed with Clarity Western ECL system (Bio-Rad). Specific proteins were detected in the dark room on the medical X-ray film (Agfa) with Agfa developer and fixer. Uncropped blot scans corresponding to all blots shown in main figures are collected in Supplementary Fig. 10.

Agarose gel electrophoresis: Different amounts/dilutions of poly(I:C) were loaded on 2% agarose gel containing ethidium bromide. After electrophoresis, poly(I:C) was visualised by UV light using Gel Doc XR + Molecular Imager (Bio-Rad) with IMAGE LAB software.

**Statistical analyses**. Sample sizes for different experiments were chosen based on the commonly used range in the field (and given in figure captions) without conducting any statistical power analysis. Histograms and scatter plots were based on at least $n \geq 300$ cells from one out of at least two experiments (as specified).

Such cell sample sizes allow us to perform manual correction of cell contours that were misidentified by an initial automated contour detection; manual corrections eliminate errors that occur inevitably in automated analysis. Sample means and s.e.m. were calculated and shown on the graphs. RT PCR time-profiles from all replicates were given in Supplementary Note.

**Code availability**. Model implementations in BIONETGEN language (BNGL), MATLAB, and SBML are provided in a ZIP archive (Supplementary Data 15).

**Data availability**. The data that support the findings of this study are included in the Supplementary Materials; the remaining data are available from the corresponding author on request.

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

## Acknowledgements

We are grateful to Professor Thomas Decker for providing the MEF Stat1$^{-/-}$ cell line. We thank Dr. Bing Tian for providing cell lines and assistance with transfection conditions. This study was supported by the National Science Centre (Poland) grants No. 2011/03/B/NZ2/00281 and 2014/14/M/NZ6/00537. A.R.B. acknowledges support from grants NIAID AI062885, UL1TR001439 and NIEHS ES006676. M. Kimmel and T.L. acknowledge support from NSF/DMS-1361411 grant. M. Kimmel acknowledges support from the National Science Centre (Poland) NN519 647840 grant.

## Author contributions

T.L., A.R.B., M.Kimmel designed the study; M.C., Z.K., W.P., S.B. designed, performed and analysed experiments; M.Kochańczyk developed software for confocal microscopy image analysis; J.J.-B. and T.L. performed mathematical modelling; K.T., J.J.-B., M.Kochańczyk analysed data; all authors contributed to writing the manuscript and approved its final form.

## Additional information

**Competing interests:** The authors declare no competing financial interests.

