## [Peer Review File · Nature Communications]

Reviewers' comments:

Reviewer #1 (Remarks to the Author):

In their paper, Czerkies et al use a combination of experimental and mathematical analysis to analyze the differences between bacterial and viral activation of innate immune network. They identify a switch-like response to polyIC that is different than a pulse like response to LPS. They argue that the switch like response is a result of auto/paracrine pathway activation through STAT1 that acts to remodel the NFkB network and reduce the strength of the negative feedback in the network. Additionally, the authors make an interesting point that cellular response heterogeneity is a result of variability in the expression of the key molecular factors (PKR, RIG-I and OAS1A) that makes a small subset of the population "sentry" cells. I found the paper interesting, original and of high significance. However, there is one key technical issue that will have to be resolved before publication.

Overall the paper uses appropriate methodologies. My major technical issue with the paper relates to an implicit assumption made by the authors. The authors assume that the amount of LPS and poly-I:C that TLR4 and RIG-I are exposed to is the same in all the cells and assign the variability to biological factors like PKR, RIG-I and OAS1A. While this is reasonable assumption for a soluble factors like LPS, an intracellular PAMP like poly-I:C that is transfected with lipofectamine is likely NOT uniformly distributed. While we never transfected poly-IC, we routinely transfect cDNA and single cell distribution are always power-law. It is likely that transfected poly-IC is similar and therefore a large amount of the variability measured in cellular response is in fact deterministic response to variable input strength in poly-I:C. To address this point the authors need to do one of two things: 1. show using fluorescently conjugated poly-I:C (those are commercially available) that cellular response variability is higher than input variance. In other words, that there is no correlation between response strength and input strength. 2. Redo the experiments that analyze IRF3 and Nfkb and relate the activation strength to actual amount of poly-IC in each cell. It is possible that this new data will not support the sentry hypothesis anymore. I think that the paper is sufficiently interesting without the cell-to-cell variability aspect. But if the authors want to talk about it, they have to show that the differences are not just an issue of transfection levels.

Minor comments:

1. I did not understand the role Gillespie simulation used by the authors. What point in the paper required addition of expression noise? This could be a matter of preference, but one could argue

that "models should be as simple as possible, but not simpler" (ref: <http://quoteinvestigator.com/2011/05/13/einstein-simple/>). Therefore either the authors simplify their model or explain better why they added additional layers of complexity. I understand why they would like to have variability in model parameters / initial conditions between cells, but not why the model itself is stochastic.

2. Role of paracrine signaling. An important point not discussed by the authors is the potential danger of positive feedback that is activated by paracrine signaling. This point is discussed for the IRF3/NfκB/STAT1 pathway in recent work (Ourthiague et al. J Leukoc Biol. 2015 Jul;98(1):119-28.) .

3. The paper will be significantly strengthened by inclusion of smFISH data on the key genes (PKR, RIG-I and OAS1A) at different timepoints after polyI:C addition in wt or STAT -/- cells. I put this as a minor point since I do not think this should be a requirement for publication. But I still wanted to mention it since this data could strengthen both points made by the authors: the remodeling of the network and the causative role cellular heterogeneity.

Reviewer #2 (Remarks to the Author):

This work focuses on the negative and positive feedback interactions among the IRF3, NF-κB and STAT1/2 pathways. The authors showed the differences between poly(I:C) and LPS with a variety of techniques. Simultaneous activation of NF-κB and IRF3 by poly(I:C) triggers synthesis of IFNβ, which activates the STAT1/2 pathway in a paracrine and autocrine manner and STAT1/2 translocate to the nucleus to upregulate expression level of RIG-I, PKR and OAS1A. Sensitization by IFNβ commits apoptosis. The manuscript has a lot of results, but it is hard to see the strong points of the present study, because most of the results in the present study has not gone beyond confirmation of already known findings in the previous study. The following points listed below need to be taken into account to improve this manuscript. It is also important to show the significance of the feedback loops in viral infection.

1. Poly(I:C) leads to a delayed and sustained activation of NF-κB and IRF3. poly(I:C) stimulates not only RIG-I, but also MDA-5 and TLR3. The authors need to show evidence that MDA-5 and TLR3 are not involved in their results, by showing the lack of their expression in the cells used, or using cells lacking MDA-5 or TLR3.

2. In Fig. 5a, the authors show that the STAT1/2 pathway induces expression of PKR and OAS1A to reduce synthesis of A20 and IκBα, leading to sustained activation of LPS signal. They, however, propose that poly(I:C) responses are sustained by the similar mechanism without showing any results. The authors need to show the responses on poly(I:C) responses.

3. In Fig. 5b and 5c, the authors show that the PKR inhibitor C16 abolished degradation of A20 and IκBα. The reviewer would like the authors to show the poly(I:C) responses in the presence of C16 treatment and discuss the results.

4. In Fig. 8, poly (I:C), but not LPS, induced apoptosis in MEF cells. The reviewer would like to see the results with viral infection instead of poly(I:C) to prove their claim that viral infection makes infected cells susceptible to apoptosis.

Response to the concerns of the Reviewers

Reviewer #1

In their paper, Czerkies et al use a combination of experimental and mathematical analysis to analyze the differences between bacterial and viral activation of innate immune network. They identify a switch-like response to polyIC that is different than a pulse like response to LPS. They argue that the switch like response is a result of auto/paracrine pathway activation through STAT1 that acts to remodel the NFkB network and reduce the strength of the negative feedback in the network. Additionally, the authors make an interesting point that cellular response heterogeneity is a result of variability in the expression of the key molecular factors (PKR, RIG-I and OAS1A) that makes a small subset of the population "sentry" cells. I found the paper interesting, original and of high significance. However, there is one key technical issue that will have to be resolved before publication. Overall the paper uses appropriate methodologies.

Response: We appreciate that the Referee found our paper interesting and methodologically sound.

My major technical issue with the paper relates to an implicit assumption made by the authors. The authors assume that the amount of LPS and poly-I:C that TLR4 and RIG-I are exposed to is the same in all the cells and assign the variability to biological factors like PKR, RIG-I and OAS1A. While this is reasonable assumption for a soluble factors like LPS, an intracellular PAMP like poly-I:C that is transfected with lipofectamine is likely NOT uniformly distributed. While we never transfected poly-IC, we routinely transfect cDNA and single cell distribution are always power-law. It is likely that transfected poly-IC is similar and therefore a large amount of the variability measured in cellular response is in fact deterministic response to variable input strength in poly-I:C. To address this point the authors need to do one of two things: 1. show using fluorescently conjugated poly-I:C (those are commercially available) that cellular response variability is higher than input variance. In other words, that there is no correlation between response strength and input strength. 2. Redo the experiments that analyze IRF3 and Nfkb and relate the activation strength to actual amount of poly-IC in each cell. It is possible that this new data will not support the sentry hypothesis anymore. I think that the paper is sufficiently interesting without the cell-to-cell variability aspect. But if the authors want to talk about it, they have to show that the differences are not just an issue of transfection levels.

Response: Regarding uneven of partition of the poly(I:C) between cells, we share the intuition of the Referee and in fact in the original model we **did not** assume that the same amount of poly(I:C) enters each cell. Instead, based on the observation of the fluorescent poly(I:C) entry process (see Supplementary Video 1), we model it

in such a way, that the distribution of poly(I:C) is quite broad (see Supplementary Note, section “System inputs”).

As suggested by the Referee, in the *revised manuscript* we quantified this distribution at 4 hr after stimulation (i.e., when most of poly(I:C) enters the cells but is not yet degraded) and compared it with the model. As shown in Supplementary Fig. 5, these two distributions have a similar shape. The coefficient of variation $c_v = \sigma/\mu$ is 0.726 for the experiment, and 0.665 for the model. Next, we performed model simulations assuming uneven (original model) and even partition of poly(I:C) to found, that, as expected by the Reviewer, the nonuniformity in poly(I:C) uptake contributes to heterogeneity of cell responses (see Supplementary Note, section “Numerical simulation protocols”, and Supplementary Fig. 6, panels f vs. e). The uneven uptake of poly(I:C) causes that the fraction of cells exhibiting strong nuclear translocation is smaller (see Supplementary Fig. 6g) and as a result coefficient of variation ($c_v = \sigma/\mu$) is higher. Nevertheless, both in the experiment and in the model the response variance is higher than input variance which assures us that heterogeneity in cell population also contributes to response variability. We refer to this analysis briefly in main text.

Minor comments:

1. I did not understand the role Gillespie simulation used by the authors. What point in the paper required addition of expression noise? This could be a matter of preference, but one could argue that "models should be as simple as possible, but not simpler" (ref: <http://quoteinvestigator.com/2011/05/13/einstein-simple/>). Therefore either the authors simplify their model or explain better why they added additional layers of complexity. I understand why they would like to have variability in model parameters / initial conditions between cells, but not why the model itself is stochastic.

Response: The approach to molecular regulatory pathways modeling in terms of stochastic Markov process is well established and there is a bulk of evidence that in eukaryotes gene expression contributes to the population heterogeneity (see for example Raj *et al.*, 2006, and Dey *et al.*, 2015).

To a some extent, Gillespie-type modelling can be simplified and the heterogeneity can be obtained by assuming that cells are characterized by different parameters, and/or have initially different levels of key proteins; however, in Tay *et al.* (2010) we demonstrated that both intrinsic noise and extrinsic noise (associated with initial proteins levels variability) shapes the NF- κ B activity (see Fig. 1i-1l, and Fig. 4a therein). Here, we employ stochastic Gillespie modelling and simulated resting cells to show that stochastic gene expression introduces heterogeneity in RIG-I and PKR levels in the time point when poly(I:C) is added. In the *revised manuscript* we show that, as expected by the Referee, this heterogeneity is in part responsible for the response variability. In Supplementary Figure 6, referred to from the Supplementary Note (section Numerical simulation protocols), we compare distributions of nuclear NF- κ B and IRF3 levels obtained in stochastic simulations and deterministic simulation with initial RIG-I and PKR levels obtained from the stochastic equilibration. Nevertheless, we prefer the full stochastic simulations, as this approach gives us initial (at time point when poly(I:C) is added) distribution of system components levels, as well further system evolution within a single formalism.

2. Role of paracrine signaling. An important point not discussed by the authors is the potential danger of positive feedback that is activated by paracrine signaling. This point is discussed for the IRF3/Nf κ B/STAT1 pathway in recent work (Ourthiague *et al. J Leukoc Biol.* 2015 Jul;98(1):119-28.).

Response: Ourthiague *et al.* (2015) analyze consequences of the positive feedback (neglected by us) in which ISGF3 (STAT1/STAT2/IRF9 complex) regulates expression of IFN β . We agree that this feedback potentially could lead to uncontrolled IFN β expression in cell population above some strength of ISGF3 mediated feedback. The IFN β storm can be avoided because (as suggested by authors) although it can bind to the same

motif as IRF3, ISGF3 may be sterically impeded from binding the AP-1- and NF- κ B-bound IFN β enhanceosome. The additional explanation can be that the negative feedback mediated by SOCS1 attenuates STAT1 activity. In Figure 7a we show that after IFN β prestimulation, STAT1 level builds up, but at some point STAT1 activity sharply drops. We attribute this to SOCS1-mediated feedback – but for sure it requires further study. In the *revised manuscript* we refer and briefly refer to this additional STAT1-mediated feedback loop.

3. The paper will be significantly strengthened by inclusion of smFISH data on the key genes (PKR, RIG-I and OAS1A) at different timepoints after polyI:C addition in wt or STAT -/- cells. I put this as a minor point since I do not think this should be a requirement for publication. But I still wanted to mention it since this data could strengthen both points made by the authors: the remodeling of the network and the causative role cellular heterogeneity.

Response: We think that the heterogeneity of cell responses is well documented at NF- κ B and IRF3 translocation and STAT1 activation level. Zhao *et al.* (2012) demonstrated in MEFs that stochastic IFN β expression is a consequence of cell-to-cell variability in the levels and/or activities of limiting components; in particular RIG-I and MAVS. We therefore choose not to supplement our experimental data by additional smFISH on RIG-I, PKR, and OAS1A genes.

Reviewer #2

This work focuses on the negative and positive feedback interactions among the IRF3, NF- κ B and STAT1/2 pathways. The authors showed the differences between poly(I:C) and LPS with a variety of techniques. Simultaneous activation of NF- κ B and IRF3 by poly(I:C) triggers synthesis of IFN β , which activates the STAT1/2 pathway in a paracrine and autocrine manner and STAT1/2 translocate to the nucleus to upregulate expression level of RIG-I, PKR and OAS1A. Sensitization by IFN β commits apoptosis. The manuscript has a lot of results, but it is hard to see the strong points of the present study, because most of the results in the present study has not gone beyond confirmation of already known findings in the previous study.

Response: Because the reference to “the previous study” is not given explicitly, we understand that the Reviewer means our previous paper:

Bertolusso, R. *et al.* Dynamic cross talk model of the epithelial innate immune response to double-stranded RNA stimulation: Coordinated dynamics emerging from cell-level noise. *PLOS One* **9**, e93396 (2014).

The Reviewer should notice that the model described in the previous paper (see Fig. 10 therein) contains neither interferon signaling nor activation of the STAT pathway. The key finding of our current study (graphically summarized in Fig. 1a and then shown in more detail in Fig. 6a) is that co-activation of IRF3 and NF- κ B pathways leads to the activation of the STAT pathway via IFN β signaling. Feedbacks from the STAT pathway to IRF3 and NF- κ B (mediated by RIG-I, PKR, and OAS1A) allows for stabilization of activity of these two transcription factors (observed at the population level, Fig. 1c, as well as single-cell level, Fig. 2a). Therefore, it seems that the Reviewer overlooked the main result of the study, which is reflected in the title and abstract of the manuscript.

The following points listed below need to be taken into account to improve this manuscript. It is also important to show the significance of the feedback loops in viral infection.

1. *Poly(I:C) leads to a delayed and sustained activation of NF- κ B and IRF3. poly(I:C) stimulates not only RIG-I, but also MDA-5 and TLR3. The authors need to show evidence that MDA-5 and TLR3 are not involved in their results, by showing the lack of their expression in the cells used, or using cells lacking MDA-5 or TLR3.*

Response: While MDA-5, TLR3 and RIG-I can all be activated by poly(I:C), they differ in subcellular localization and ligand specificity. TLR3 is unlike to interface with free cytoplasmic poly(I:C) after transfection with a lipophilic agent, such as Lipofectamine used in our experiments, because it localizes to the ER, endosomal and plasma membranes (McGettrick & O'Neill, 2010; Pohar *et al.*, 2014).

MDA-5 and RIG-I have been shown to preferentially recognize poly(I:C) molecules of different sizes (Kato *et al.*, 2008), with RIG-I favoring the kind of short (0.1–1 kb) molecules that were used in our experiments (see Supplementary Data 11). Consequently, RIG-I appears as the most likely receptor for the poly(I:C) stimulation used in our experimental set-up.

Furthermore, as MDA-5 and RIG-I each associate with adaptor protein MAVS (Reikine *et al.*, 2014) and their signals converge downstream at TRAF3 (Häcker *et al.*, 2005; Oganessian *et al.*, 2006; Saha *et al.*, 2006), within the scope of our model the two receptors together can be thought of as one receptor pool.

Considering that the expression levels of MDA-5 and RIG-I in resting (bronchial epithelial) cells are similar (Slater *et al.*, 2010), a many-fold increase in RIG-I expression after poly(I:C) stimulation (see Figure 4a, protein, and 4b mRNA) guarantees the increase of the entire pool regardless of contribution from MDA-5.

In the revised text we briefly explain why we restrict ourselves to RIG-I receptor.

2. and 3. *In Fig. 5a, the authors show that the STAT1/2 pathway induces expression of PKR and OAS1A to reduce synthesis of A20 and I κ B α , leading to sustained activation of LPS signal. They, however, propose that poly(I:C) responses are sustained by the similar mechanism without showing any results. The authors need to show the responses on poly(I:C) responses. In Fig. 5b and 5c, the authors show that the PKR inhibitor C16 abolished degradation of A20 and I κ B α . The reviewer would like the authors to show the poly(I:C) responses in the presence of C16 treatment and discuss the results. 4. In Fig. 8, poly(I:C), but not LPS, induced apoptosis in MEF cells.*

Response: The Reviewer should notice that in Fig. 5a we show system responses to LPS for which STAT1/2 pathways is not activated. The responses to poly(I:C) with and without PKR inhibitor C16 are shown in Fig. 5b and 5c. This is, the Reviewer asks us to show results that are actually shown in mentioned Figs. 5b and 5c.

The sustained responses of system components to poly(I:C) are shown in Figs 2a,b,c (NF- κ B activity profiles), Fig. 4 (WT compared with STAT1^{-/-}), Fig. 7 where the effect of interferon pre-stimulation on poly(I:C) stimulation is analyzed. It seems that Reviewer overlooked these three figures reporting key results of the manuscript.

4. *The reviewer would like to see the results with viral infection instead of poly(I:C) to prove their claim that viral infection makes infected cells susceptible to apoptosis.*

Response: In this study, we use poly(I:C), an established viral analog to elucidate the general mechanisms controlling the decision making process that underlines innate immune response at the single cell and population level. We are not claiming that viral infection in general makes infected cells susceptible to

apoptosis; some viruses trigger apoptosis, some do not. The viruses use multiple mechanism to compromise immune system and therefore the responses to particular viruses can substantially differ. Nevertheless, system level analysis of responses to poly(I:C) is in our opinion the first step to decipher responses to particular viruses.

References

- Ashall, L., *et al.* Pulsatile stimulation determines timing and specificity of NF- κ B-dependent transcription. *Science* **324**, 242–6 (2009).
- Bertolusso, R., *et al.* Dynamic cross talk model of the epithelial innate immune response to double-stranded RNA stimulation: Coordinated dynamics emerging from cell-level noise. *PLoS One* **9**(4), e93396 (2014).
- Burman, A., *et al.* The role of leukocyte-stromal interactions in chronic inflammatory joint disease. *Joint Bone Spine* **72**, 10–16 (2005).
- Cheng, Z., *et al.* Distinct single-cell signaling characteristics are conferred by the MyD88 and TRIF pathways during TLR4 activation. *Immunology* **8**, ra69 (2015).
- Covert, M.W., *et al.* Achieving stability of lipopolysaccharide-induced NF-kappaB activation. *Science* **309**, 1854–7 (2005).
- Dey, S. S., *et al.* Orthogonal control of expression mean and variance by epigenetic features at different genomic loci. *Mol. Syst. Biol.* **11**, 806 (2015).
- Häcker, H., *et al.* Specificity in Toll-like receptor signalling through distinct effector functions of TRAF3 and TRAF6. *Nature* **439**, 204–7 (2005).
- Hess, C. B., *et al.* The induction of interferon production in fibroblasts by invasive bacteria: a comparison of Salmonella and Shigella species. *Microb. Pathog.* **2**, 111–20 (1989).
- Herdy, B., *et al.* Translational control of the activation of transcription factor NF- κ B and production of type I interferon by phosphorylation of the translation factor eIF4E. *Nat. Immunol.* **13**, 543–50 (2012).
- Hoffmann, A., *et al.* The IkappaB-NF-kappaB signaling module: temporal control and selective gene activation. *Science* **298**, 1241–5 (2002).
- Kato, H., *et al.* Length-dependent recognition of double-stranded ribonucleic acids by retinoic acid-inducible gene-I and melanoma differentiation-associated gene 5. *J. Exp. Med.* **205**, 1601–10 (2008).
- Kellogg, R. A., *et al.* Digital signaling decouples activation probability and population heterogeneity. *eLife* **4**, e08931 (2015).
- Kurt-Jones, E. A., *et al.* Use of murine embryonic fibroblasts to define Toll-like receptor activation and specificity. *J. Endotoxin Res.* **10**, 419–24 (2004).
- Lee, T. K., *et al.* A noisy paracrine signal determines the cellular NF-kappaB response to lipopolysaccharide. *Sci. Signal.* **2**, ra65 (2009).
- Lipniacki, T., *et al.* Mathematical model of NF- κ B regulatory module. *J. Theor. Biol.* **228**, 195–215 (2004).
- Lipniacki, T., *et al.* Single TNF α trimers mediating NF- κ B activation: Stochastic robustness of NF- κ B signaling. *BMC Bioinformatics* **8**:376 (2007).
- Loo, Y.M., *et al.* Distinct RIG-I and MDA5 signaling by RNA viruses in innate immunity. *J. Virol.* **82**, 335–45 (2008).
- McGettrick, A. F., & O'Neill, L. A. J. Localisation and trafficking of Toll-like receptors: an important mode of regulation. *Curr. Opin. Immunol.* **22**, 1, 20–7 (2010).
- Nelson, D. E., *et al.* Oscillations in NF- κ B signaling control the dynamics of gene expression. *Science* **306**, 704–8 (2004).
- Oganesyan, G., *et al.* Critical role of TRAF3 in the Toll-like receptor-dependent and -independent antiviral response. *Nature* **439**, 208–11 (2006).
- Otte, J. M., *et al.* Intestinal Myofibroblasts in Innate Immune Responses of the Intestine. *Gastroenterology* **124**, 1866–78 (2003).
- Ourthiague, D. R., *et al.* Limited specificity of IRF3 and ISGF3 in the transcriptional innate-immune response to double-stranded RNA. *J. Leukoc. Biol.* **98**:119–28 (2015).
- Pohar, J., *et al.* The Ectodomain of TLR3 Receptor Is Required for Its Plasma Membrane Translocation. *PLoS ONE* **9**, e92391 (2014).
- Powell, D. W., *et al.* Mesenchymal Cells of the Intestinal Lamina Propria. *Annu. Rev. Physiol.* **73**, 213–37 (2011).
- Raj, A., *et al.* Stochastic mRNA synthesis in mammalian cells. *PLoS Biol.* **4**, e309 (2006).
- Rand, U., *et al.* Multi-layered stochasticity and paracrine signal propagation shape the type-I interferon response. *Mol. Syst. Biol.* **8**, 584 (2012).
- Reikine, S., *et al.* Pattern recognition and signaling mechanisms of RIG-I and MDA5. *Front. Immunol.* **5**, 342 (2014).
- Saha, S. K., *et al.* Regulation of antiviral responses by a direct and specific interaction between TRAF3 and Cardif. *EMBO J.* **25**, 3257–63, (2006).
- Slater, L., *et al.* Co-ordinated role of TLR3, RIG-I and MDA5 in the innate response to rhinovirus in bronchial epithelium. *PLoS Pathog.* **6**, e1001178, (2010).
- Tay, S., *et al.* Single-cell NF-kappaB dynamics reveal digital activation and analogue information processing. *Nature* **466**, 267–71 (2010).
- Van Linthout, S., *et al.* Crosstalk between fibroblasts and inflammatory cells. *Cardiovasc. Res.* **102**, 258–69 (2014).
- Zhao, M., *et al.* Stochastic expression of the interferon- β gene. *PLoS Biol.* **10**, e1001249, (2012).

Reviewers' comments:

Reviewer #1 (Remarks to the Author):

I felt that some of the points addressed nicely in the rebuttal were pretty hard to find in the revised main text. Specifically, the point related to the uneven poly I:C distribution is a completely experimental / biological concern and has nothing to do with the model. Potentially readers that are not concerned with the details of the model will still interpret these results. And it is important that it is clear to them that a big component of the observed variability is related to uneven input distribution. Yet the authors chose to address this point only under the section related to details of the model. I would strongly recommend that these sections and the reference to supp figures 5 and 6 would be moved to more prominent place. Similarly, although the rebuttal letter mentions that main text was revised to discuss relationship to previous work on positive feedback, it was not clear to me what actual changes were made.

Reviewer #2 (Remarks to the Author):

The reviewer still does not understand the strong point of this manuscript over the previous enormous studies on type I IFN signaling from the many groups other than the authors's. The manuscript is still hard to understand despite a large amount of results.

It is possible that lipofectamine facilitates polyIC delivery not only to the cytoplasm but also to endolysosomes. It is important to experimentally exclude the involvement of TLR3.

Response to the Reviewers' comments

We thank the reviewers for their time and comments on our study. In response, we have conducted additional experiments on the *Tlr3*^{-/-} MEFs that demonstrate that TLR3 signaling does not substantially contribute to the studied signaling system in fibroblasts in response to poly(I:C). We also modified manuscript to more accurately address issues mentioned by the first Reviewer. Below we provide point-by-point responses to the Reviewers' concerns and refer to the revisions made in the revised text in response to these concerns.

Reviewer #1

Comment: I felt that some of the points addressed nicely in the rebuttal were pretty hard to find in the revised main text. Specifically, the point related to the uneven poly I:C distribution is a completely experimental/biological concern and has nothing to do with the model. Potentially readers that are not concerned with the details of the model will still interpret these results. And it is important that it is clear to them that a big component of the observed variability is related to uneven input distribution. Yet the authors chose to address this point only under the section related to details of the model. I would strongly recommend that these sections and the reference to supp figures 5 and 6 would be moved to more prominent place.

Response: We agree with the Referee that uneven poly I:C distribution is a completely experimental concern and should not be introduced in the Model subsection. In the Revised manuscript we discuss this issue in the last paragraph before the Model subsection, where we also refer to Supplementary Figures 3 and 4 (previously Supp. Figs. 5 and 6). We quote this paragraph below.

“Single-cell data show that responses to poly(I:C) are heterogeneous (Fig. 2). There are at least two potential sources of this heterogeneity: uneven partition of poly(I:C) between cells due to lipofectamine-based delivery and initial variability of key proteins levels. Analysis of cellular uptake of fluorescent poly(I:C) (Supplementary Video 1) indicates a broad distribution of cellular poly(I:C) at 4 hr after stimulation, i.e., when most of poly(I:C) enters cells but is not yet degraded (Supplementary Fig. 3). The uneven uptake of poly(I:C) causes that the fraction of cells exhibiting strong nuclear translocation is smaller. Both experiment and the model indicate that the response variance (at the level of NF- κ B and IRF3 activation) is higher than the input variance (at the level of cellular poly(I:C) distribution), which indicates that heterogeneity in initial cell states also contributes to response variability (Supplementary Fig. 4).”

Comment: Similarly, although the rebuttal letter mentions that main text was revised to discuss relationship to previous work on positive feedback, it was not clear to me what actual changes were made.

Response: We moved the reference to the previous work on the positive feedback to the second paragraph before the Model subsection. We paste the relevant sentences below.

“Recently, Ourthiague *et al.* (2015)³² analysed consequences of another positive feedback, in which ISGF3 (STAT1/STAT2/IRF9 complex) regulates expression of IFN β . This paracrine feedback could potentially lead to an uncontrolled IFN β expression in cell population. The IFN β storm could be avoided if ISGF3 complex,

which binds the same motif as IRF3, due to its size may be sterically impeded by AP-1 or NF- κ B from binding to the IFN β enhanceosome.”

Reviewer #2

Comment: The reviewer still does not understand the strong point of this manuscript over the previous enormous studies on type I IFN signaling from the many groups other than the authors's. The manuscript is still hard to understand despite a large amount of results.

Response: We concur that there exists an enormous literature on type I IFN signalling. Yet these discoveries have never been quantitatively reconciled with a comprehensive mathematical model that analyzes intertwined feedbacks coupling the NF- κ B, IRF, STAT, and IFN signalling. Neither are we aware of such a model, nor is the Reviewer referring us to one. The most closely related model was developed by Rand *et al.* (ref. 17), but it is based on a very limited data set, and the entire regulatory process is described in a highly simplified way via stochastic transitions between nine distinguished states of the cell (Rand *et al.*, Fig. 6a).

What distinguishes our contribution from existing work is that it proposes a model that delineates the mechanisms of cell fate decision-making in the innate immune response. The model is based on a comprehensive set of data gathered by us on WT MEFs, as well as *RelA*, *STAT1*, and *TLR3*-deficient MEF lines, both at population and single cell level. Some of experimental results were expected based on data on other cell lines, nevertheless these experiments had to be performed in order to achieve a complete picture of crosstalk between the IRF, NF- κ B and INF β -JAK/STAT pathways in fibroblasts. A computational model of a signalling pathway cannot be developed by combining data collected on various cell lines that exhibit different responses, or even similar responses with different kinetics.

Comment: It is possible that lipofectamine facilitates polyIC delivery not only to the cytoplasm but also to endolysosomes. It is important to experimentally exclude the involvement of TLR3.

Response: As requested by the Reviewer, we compared responses of TLR3 KO and WT MEFs. We found almost no difference in the levels of activation of all three key transcription factors: NF- κ B, IRF3, and STAT by poly(I:C), as well as levels of PKR, OAS1A, and RIG-I proteins (see Supplementary Figure 9b,c,e). Only the apoptotic fraction was found somewhat lower in TLR3 KO MEFs (see Supplementary Figure 9e).

This, together with data showing that MDA-5 and RIG-I preferentially recognize poly(I:C) molecules of different sizes (Kato *et al.*, 2008, ref. 24), with RIG-I favoring short (0.1–1 kb) molecules the kind used in our experiments (see Supplementary Data 12), allow us to consider RIG-I as a primary signal mediator in our experimental set-up. Of note, this is also in agreement with the observation of Kato *et al.* (2008; ref. 24), that RIG-I knockout in MEFs suppresses IFN β secretion in response to short poly(I:C) chains. Our new data are mentioned for the first time in the second paragraph of subsection “Divergent NF- κ B/IRF3 activation kinetics in response to LPS and poly(I:C) stimulation” in Results, where we write:

“Poly(I:C) (delivered intracellularly by lipofectamine transfection) binds cytosolic RIG-I and MDA5 receptors^{24,25}, an event that leads to a delayed but longer-lasting activity of NF- κ B and IRF3 (Fig. 1b,c). In same cell lines Poly(I:C) activates innate immune signalling through the TLR3 receptor, located primarily in the endosomes²⁶. We found, however, that in the case when poly(I:C) is delivered by lipofectamine transfection, WT and *Tlr3*^{-/-} MEFs exhibit similar responses (see Methods).”

Then, Supplementary Fig. 9 showing the results on *Tlr3*^{-/-} MEFs are referred to in second subsection of Methods, were we wrote:

“As it is possible that Lipofectamine facilitates poly(I:C) delivery not only to the cytoplasm, where it activates RIG-I (and/or MDA5), but also to endosomes, where it can activate TLR3, it is important to determine the (potential) contribution of TLR3 in innate immune signalling. We thus verified that poly(I:C) (delivered by lipid-based transfection) induces the same activation of transcription factors NF-κB, IRF3, and STAT1 (Supplementary Fig. 9b,c), and leads to the same increase of protein levels of RIG-I, PKR, and OAS1A (Supplementary Fig. 9d) in WT and *Tlr3*^{-/-} MEFs. The apoptotic rates after poly(I:C) either with or without IFNβ prestimulation were somewhat lower for *Tlr3*^{-/-} cells (Supplementary Fig. 9e), whereas the fraction of cells showing activation of NF-κB was somewhat lower for WT cells. Moreover, our estimates showed poly(I:C) length to be within the 100–1000 bp range (see Supplementary Data File 12), corresponding to the molecular mass range of 67–670 kDa. Poly(I:C) of that length is preferentially bound by RIG-I, while MDA5 is known to bind longer chains²⁷. IFNβ secretion was suppressed in *Rig-I*^{-/-} MEFs in response to short-chain poly(I:C) stimulation²⁷.”

Atypically, we decided to refer to Supplementary Figure 9 in Methods, as it is relevant to the specific method of lipid-based poly(I:C) transfection, and, additionally, because it is more convenient to discuss TLR3 KO responses after presenting results on WT cells, not in the section where we first mention that RIG-I is the primary receptor for the chosen cell line and method of poly(I:C) stimulation.